# Inhibition enhances spatially-specific calcium encoding of synaptic input patterns in a biologically constrained model

Daniel B Dorman[1], Joanna Jędrzejewska-Szmek[2†], Kim T Blackwell[3†*]

[1]Interdisciplinary Program in Neuroscience, George Mason University, Fairfax, United States; [2]Krasnow Institute for Advanced Study, George Mason University, Fairfax, United States; [3]Interdisciplinary Program in Neuroscience, Bioengineering Department, Krasnow Institute for Advanced Study, George Mason University, Fairfax, United States

**Abstract** Synaptic plasticity, which underlies learning and memory, depends on calcium elevation in neurons, but the precise relationship between calcium and spatiotemporal patterns of synaptic inputs is unclear. Here, we develop a biologically realistic computational model of striatal spiny projection neurons with sophisticated calcium dynamics, based on data from rodents of both sexes, to investigate how spatiotemporally clustered and distributed excitatory and inhibitory inputs affect spine calcium. We demonstrate that coordinated excitatory synaptic inputs evoke enhanced calcium elevation specific to stimulated spines, with lower but physiologically relevant calcium elevation in nearby non-stimulated spines. Results further show a novel and important function of inhibition—to enhance the difference in calcium between stimulated and non-stimulated spines. These findings suggest that spine calcium dynamics encode synaptic input patterns and may serve as a signal for both stimulus-specific potentiation and heterosynaptic depression, maintaining balanced activity in a dendritic branch while inducing pattern-specific plasticity.
DOI: https://doi.org/10.7554/eLife.38588.001

*For correspondence:
kblackw1@gmu.edu

Present address: †Department of Neurophysiology, Nencki Institute of Experimental Biology, Warsaw, Poland

Competing interests: The authors declare that no competing interests exist.

## Introduction

Neurons receive information from other neural cells in the form of patterns of activation of different synaptic inputs. Such input patterns differ in their locations on the dendritic tree (spatial properties) and their timing (temporal properties) (*London and Häusser, 2005*). As a given neuron may receive synaptic inputs from hundreds to thousands of other neurons, a critical question in neuroscience is *how* multiple synaptic inputs are integrated to produce neuronal output. Further, certain patterns of input can induce synaptic plasticity—neural activity-dependent changes in synaptic efficacy that underlie learning and memory. Yet, it remains unclear how spatiotemporal properties of synaptic input patterns may affect synaptic plasticity (*Destexhe and Marder, 2004*; *van Bommel and Mikhaylova, 2016*).

Dendrites are capable of complex, non-linear forms of synaptic integration, which are sensitive to the spatiotemporal properties of synaptic inputs (*Stuart and Spruston, 2015*). For instance, in vitro studies have shown that near-simultaneous stimulation of a group of spatially clustered excitatory synapses on a thin dendritic branch can elicit supralinear, prolonged membrane depolarizations in the soma (known as plateau potentials). These plateau potentials have been observed in pyramidal neurons of the cortex (*Larkum et al., 2009*; *Schiller et al., 2000*) and hippocampus (*Golding et al., 2002*; *Harnett et al., 2012*; *Makara and Magee, 2013*), and also in spiny projection neurons of the

**eLife digest** How do we form new memories? The human brain contains almost 90 billion neurons, which communicate with one another at junctions called synapses. Each neuron has a shape a little like that of a tree, and is covered in branches called dendrites. Synapses typically form between the end of one neuron and a dendrite on another. Most scientists believe that the brain forms new memories by changing the strength of these synapses. But a number of questions remain about how this process works.

There are two types of synapses: excitatory and inhibitory. When an excitatory synapse becomes active, calcium ions flow into the dendrite of the receiving neuron. The calcium ions then trigger processes inside the cell that are essential for changing the strength of the synapse, and thus forming a memory. But what happens when an inhibitory synapse becomes active? How does this affect memory?

Additionally, each neuron forms synapses with thousands of others, with several synapses on a single dendrite. To form a memory about a specific experience, the brain must strengthen only the synapses that relate to that experience. How does the brain manage to target these synapses specifically? Do the synapses need to be clustered on the same dendritic branch, or can they be spread apart? And do all the synapses need to be active at exactly the same time?

Dorman et al. investigated these questions by developing a computer model of a neuron. Testing the model revealed that the synapses related to an experience do not all need to be active at exactly the same time to form a memory. Moreover, the synapses can be spread across multiple dendrites. Finally, the model showed that inhibitory synapses are critical for preventing calcium ions from spreading within dendritic branches and entering inactive synapses. This ensures that only the synapses active during a specific experience become stronger.

Many brain disorders, including substance abuse and addiction, involve errors in the processes that underlie learning and memory. By increasing our understanding of how the structure of brain cells supports these processes, the current findings could one day lead to better treatments for these and other disorders.

DOI: https://doi.org/10.7554/eLife.38588.002

striatum (*Du et al., 2017*; *Mahfooz et al., 2016*; *Oikonomou et al., 2014*; *Plotkin et al., 2011*). These non-linear responses to spatiotemporally clustered synaptic input can induce synaptic plasticity. Specifically, long-term potentiation (LTP) of synaptic inputs can be induced by stimulation of clustered synapses, independently of postsynaptic action potentials (*Brandalise et al., 2016*; *Golding et al., 2002*; *Gordon et al., 2006*; *Losonczy et al., 2008*).

Calcium influx into neuronal dendrites and spines is a critical mechanism linking synaptic input patterns to synaptic plasticity, as calcium is required for most forms of neuronal plasticity throughout the brain (*Greer and Greenberg, 2008*; *Higley and Sabatini, 2008*; *Zucker, 1999*). The conjunction of synaptic inputs and postsynaptic depolarization produces calcium influx through the NMDA subtype of glutamate receptor (NMDAR) channels (*Bartol et al., 2015*; *Schiller et al., 1998*; *Sjöström and Nelson, 2002*). Activation of calcium-permeable ligand-gated or voltage-gated ion channels also yields calcium influx. The resulting elevation in intracellular calcium acts as a second messenger to initiate multiple signaling cascades that produce various forms of synaptic plasticity. Calcium, therefore, connects the electrical activity at the network or neuronal level to the subcellular level of biochemical signaling and plasticity. The relationship between calcium and plasticity is complex, as calcium elevation is required for both LTP and long-term depression (LTD). Both experiments and theory propose that plasticity outcomes depend on the specific dynamics of intracellular calcium, including amplitude, duration, and location (*Evans and Blackwell, 2015*; *Graupner and Brunel, 2012*). Thus, determining how calcium dynamics in dendrites and spines depend on spatiotemporal patterns of synaptic input will advance our understanding of how those same patterns induce plasticity and ultimately influence learning and memory.

Spatiotemporally clustered synaptic inputs that produce supralinear plateau potentials (also called NMDA spikes) also cause elevated dendritic calcium concentration localized to the stimulated dendritic branch (*Antic et al., 2010*; *Larkum et al., 2009*; *Major et al., 2008*; *Schiller et al., 2000*). In

vivo, NMDAR-dependent calcium transients that are limited to specific dendritic branches and spines of pyramidal neurons correspond with spine-specific structural plasticity and behavioral learning (*Cichon and Gan, 2015*). In vitro, repeated synaptic stimulation of neighboring spines can result in supralinear spine calcium transients and LTP (*Weber et al., 2016*), even in the absence of somatic plateau potentials. Thus, spatiotemporally clustered patterns of synaptic inputs are critical for information processing and plasticity, but it is unclear how distributed input patterns, which likely occur frequently in vivo, produce non-linear synaptic responses and affect synaptic plasticity.

Although much of the literature has focused on synaptic integration in pyramidal neurons, NMDAR-dependent plateau potentials also have been observed in the spiny projection neurons (SPNs) of the striatum, which is the input nucleus of the basal ganglia (*Du et al., 2017*; *Mahfooz et al., 2016*; *Oikonomou et al., 2014*; *Plotkin et al., 2011*). The striatum integrates glutamatergic input from cortex and thalamus and dopaminergic input from substantia nigra to learn goal-directed actions, motor skills, and habits (*Kreitzer and Malenka, 2008*). Calcium elevation and dopamine are required for synaptic plasticity in SPNs (*Yagishita et al., 2014*). It has been suggested that calcium elevation (through downstream signaling events) may generate a 'synaptic eligibility trace' that, when followed by dopamine stimulation, produces LTP (*Shindou et al., 2018*); thus calcium dynamics are critical even in brain regions that require dopamine for synaptic plasticity. Similar to pyramidal neurons, near-synchronous synaptic input to a cluster of 10 – 20 neighboring spines evokes NMDAR-dependent plateau potentials in SPNs (*Du et al., 2017*; *Plotkin et al., 2011*), but only when the cluster of spines is located distally, and not proximally. Although not yet demonstrated, this supralinearity may produce synaptic plasticity at these distal SPN synapses.

SPNs and pyramidal neurons exhibit key differences which motivate the further study of SPNs. In contrast to pyramidal cells, SPNs lack the morphologically distinct apical, oblique, and basal dendritic branches characteristic of pyramidal neurons (*Gertler et al., 2008*; *Spruston, 2008*). Whereas pyramidal neurons exhibit sodium- and calcium-spikes in addition to NMDA spikes (*Stuart and Spruston, 2015*), SPNs lack sodium channels in distal dendrites and dendritic calcium-spikes have not been measured in these neurons (*Day et al., 2008*; *Plotkin et al., 2011*). SPNs rest at more hyperpolarized membrane potentials than pyramidal neurons, in part because of differences in hyperpolarization-activated ion channels—SPNs strongly express inward rectifying potassium channels (KIR) and lack the hyperpolarization-activated cyclic nucleotide-gated (HCN) channels seen in pyramidal neurons (*Nisenbaum and Wilson, 1995*). SPNs also transition between hyperpolarized down-states and depolarized up-states in vivo (*Wilson and Kawaguchi, 1996*), and dendritic non-linearities may play an important role in driving these state transitions (*Du et al., 2017*; *Plotkin et al., 2011*). Lastly, cortical axons make only 1 – 3 synapses with a single SPN (*Kincaid et al., 1998*), and the large number of synaptic inputs required to produce an upstate (*Blackwell et al., 2003*; *Stern et al., 1998*) suggests that in SPNs spatiotemporally dispersed synaptic inputs occur frequently.

The striatal microcircuit also differs significantly from that of the cortex or hippocampus. The local circuit of the striatum is almost entirely inhibitory (*Burke et al., 2017*), including the SPN collaterals. Other sources of inhibition include fast spiking interneurons (FSIs), low-threshold spiking interneurons (LTSIs), and neurogliaform (NGF) interneurons (*Burke et al., 2017*; *Ibáñez-Sandoval et al., 2011*; *Kawaguchi et al., 1995*; *Koos et al., 2004*; *Straub et al., 2016*; *Tepper and Bolam, 2004*; *Koós and Tepper, 1999*). Synaptic inputs onto SPNs from these sources exhibit distinct spatial and temporal properties. FSIs fire at high rates and are limited to the soma and proximal dendrites of SPNs. In contrast, SPN collaterals, LTSIs, and NGFs target distal dendrites of SPNs, and NGF synapses further display distinctly slow GABA$_A$ kinetics. These distinct sources of inhibition have been shown to regulate plateau potentials in somatic recordings (*Du et al., 2017*). Yet, although inhibition is clearly important for striatal function, its role in regulating local calcium transients in dendritic spines is unknown.

When clustered synaptic input produces a plateau potential, it is unclear how synapse-specificity is maintained. Calcium imaging experiments indicate that the entire dendritic branch at the site of clustered synaptic input experiences robust calcium elevation (*Plotkin et al., 2011*), and the plateau potential likely propagates from the dendritic branch into neighboring non-stimulated dendritic spines (*Koch and Zador, 1993*), which could minimize synapse-specificity. However, neither experiments nor models have investigated the degree of synapse-specificity in the calcium response during supralinear plateau potentials, in any neuron type.

Given the importance of synaptic activity patterns for information processing and plasticity, understanding the role of both spatiotemporally clustered and distributed synaptic input patterns on calcium dynamics is critical. However, because of experimental technical constraints, computational modeling is required to investigate the response to spatiotemporally dispersed inputs. We address these critical issues regarding the effect of spatiotemporal input patterns on calcium dynamics in stimulated spines, non-stimulated spines, and dendritic branches in a detailed computational SPN model. We show that both dispersed and clustered synaptic inputs can evoke supralinear calcium influx into stimulated spines with spatial specificity. Lastly, our most novel finding is that inhibition enhances spatial- and synapse-specificity of spine calcium transients during plateau potentials.

## Results

### Multiscale model reproduces electrophysiology and calcium-imaging experiments

We developed a detailed biophysical SPN model to investigate the effect of spatially and temporally clustered and distributed synaptic inputs on spine calcium dynamics (*Figure 1A*). Ion channel densities were tuned to reproduce SPN electrophysiological characteristics in response to current injection (*Figure 1B*). The model exhibits the inward rectification and sag in response to hyperpolarizing current injection, latency to first action potential, shallow AHP amplitude, input resistance, and low firing frequency characteristic of SPN recordings (*Nisenbaum and Wilson, 1995*).

Calcium dynamics were incorporated into the model to reproduce an array of experimental data (*Figure 1A,C*). Voltage compartments were subdivided into smaller calcium compartments—either radial diffusion shells in the soma and dendrites, or axial diffusion slabs in the spines. Each shell or slab also had diffusible calcium buffers (calmodulin and calbindin), a low affinity fixed buffer (*Matthews and Dietrich, 2015*), and plasma membrane calcium pumps (for compartments adjacent to membranes). Maximal conductances of voltage-gated calcium channels (VGCCs) were tuned to reproduce experiments measuring the calcium concentration elevations in dendrites and spines in response to a back-propagating action potential (bAP). Conductances of NMDAR and AMPAR channels were tuned to reproduce calcium imaging experiments showing spine calcium elevation during a single excitatory postsynaptic potential (EPSP). *Figure 1C* shows that bAP-evoked calcium elevation is greater in proximal dendrites than in the soma and decreases with distance in tertiary dendrites, consistent with experimental reports (*Day et al., 2008*; *Kerr and Plenz, 2002*). Also, peak calcium in a proximal spine from a bAP was 0.18 µM with a time constant of decay of 74 ms, and in response to a single EPSP peaked at 0.2 µM with a time constant of decay of 73 ms, similar to experimental results when simulated under similar calcium-indicator conditions (*Shindou et al., 2011*). The relative contributions of specific VGCC types and synaptic calcium sources to spine calcium elevation were also tuned to reproduce experimental data (*Carter and Sabatini, 2004*; *Higley and Sabatini, 2010*). Blockade of NMDARs, AMPARs, and T-type, R-type, or L-type VGCCs reduced the spine calcium elevation in response to a single EPSP. Spine calcium elevation in response to a bAP or EPSP was evaluated for parameter variations of ±10% (*Figure 1—figure supplement 1*). These results demonstrate that the model peak spine calcium response is robust to parameter variations, with at most a 10% change in response to a bAP and a 30% change in response to an EPSP (within reported experimental variability) (*Shindou et al., 2011*). To further assess model robustness, we evaluated the fates of calcium entering a dendritic spine during a single EPSP for 100 ms following synaptic stimulation. Calcium fates, which were calculated as the quantity (moles) of free calcium, buffered calcium, pumped calcium, and diffused calcium per timestep (*Figure 1—figure supplement 2*), exhibited similar dynamics to a published computational model with three-dimensional reaction-diffusion in reconstructed dendritic spines from pyramidal neurons (*Bartol et al., 2015*). Together, the ability to reproduce multiple sources of both electrophysiology and calcium-imaging data suggest that the model is well-suited to investigate the effects of synaptic activity on calcium dynamics.

### Synaptic stimulation of distally located spines produces non-linear spine calcium transients

SPNs exhibit plateau potentials in response to spatiotemporally clustered synaptic inputs to distal dendritic spines, but it is unclear how these plateau potentials affect spine calcium dynamics or

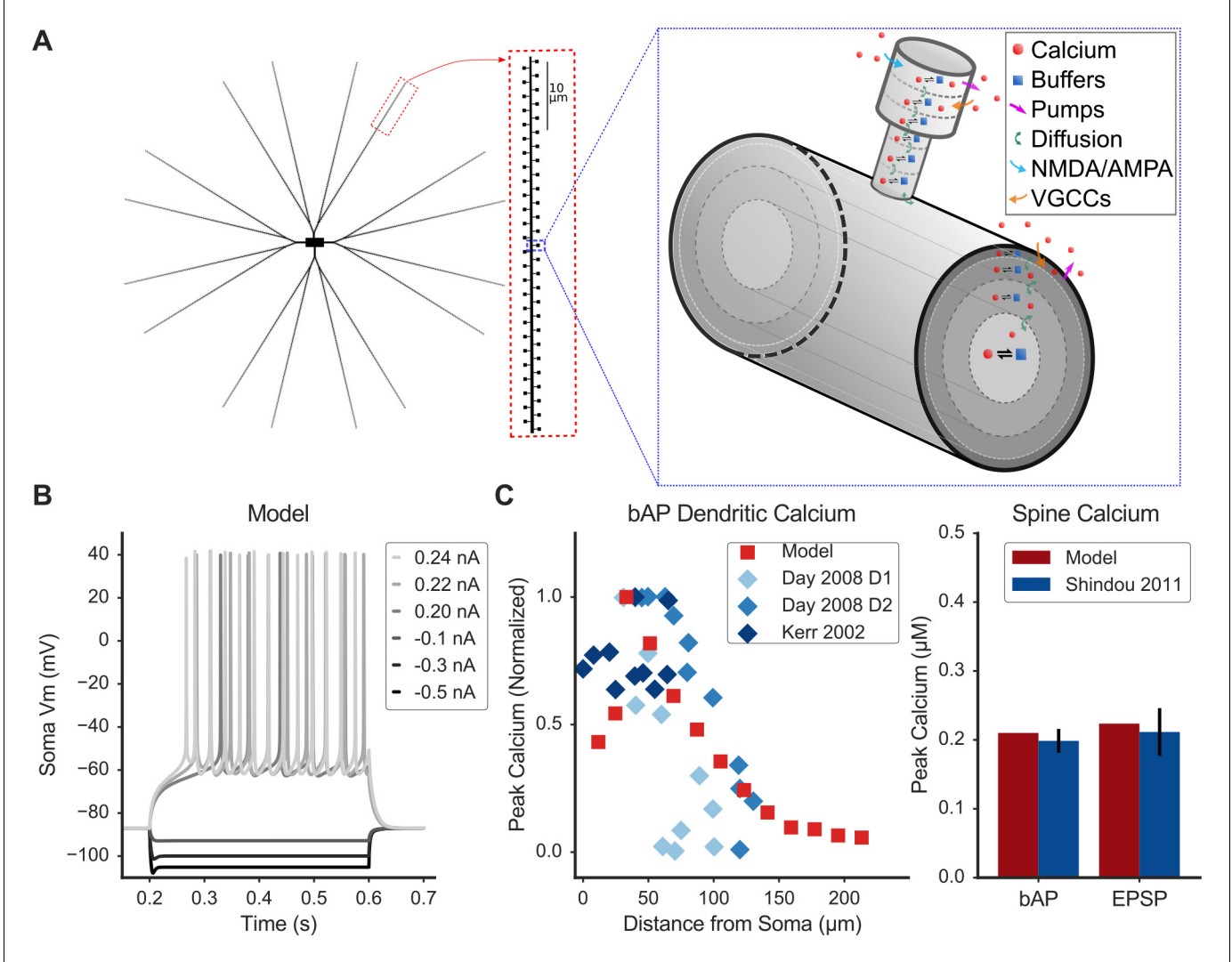

**Figure 1.** Spiny Projection Neuron model reproduces electrophysiology and calcium imaging data. (**A**) Model schematic of morphology (left) and calcium dynamics in spine and dendrite (right). Characteristic morphology with dendritic spines is based on estimated values from morphological reconstructions, including tapered dendrites. The model contains ionic and synaptic channels (not depicted). Right: a single spine and parent dendritic compartment showing axial diffusion layers in the spine head and neck and radial diffusion shells in the dendrite. The model includes sophisticated Ca dynamics (red circles indicate calcium ions): diffusion (green arrows), buffers (blue squares - calmodulin, calbindin, fixed buffer; reversible reaction with calcium indicated by black reaction arrows), pump extrusion (purple arrows), and influx via voltage-gated calcium channels (VGCCs; orange arrows) and synaptic (NMDA/AMPA) channels (blue arrows). (**B**) Model exhibits similar response to electrophysiological current injection steps, including characteristic sag following hyperpolarizing current, latency to first action potential, firing rate, and AHP shape (*Nisenbaum and Wilson, 1995*). (**C**) Model calcium dynamics (red) are consistent with experiments (blue) for dendritic calcium (left) vs. distance from soma in response to a back-propagating action potential (bAP) and for spine calcium (right) in response to a bAP or synaptic stimulation (EPSP). Legend entries refer to published experimental calcium imaging data; D1 or D2 refers to dopamine receptor expression of identified SPNs: (*Day et al., 2008*; *Kerr and Plenz, 2002*; *Shindou et al., 2011*). *Figure 1—figure supplement 1* shows peak spine calcium sensitivity to parameter variations; *Figure 1—figure supplement 2* shows the fate of calcium entering a spine during an EPSP.

DOI: https://doi.org/10.7554/eLife.38588.003

The following figure supplements are available for figure 1:

**Figure supplement 1.** Parameter sensitivity for peak spine calcium concentration in response to a single bAP or EPSP.

DOI: https://doi.org/10.7554/eLife.38588.004

**Figure supplement 2.** Fate of calcium entering a spine during an EPSP.

DOI: https://doi.org/10.7554/eLife.38588.005

synaptic plasticity. Further, the likelihood of spatial clustering of synaptic inputs occurring in vivo is unknown, and may be low, as individual cortical axons make few synaptic connections to a single SPN (*Kincaid et al., 1998*). The ability of spatially distributed synaptic inputs to produce a plateau potential would increase the biological relevance of plateau potentials for in vivo function of SPNs. To investigate the role of spatial input patterns, we first verified that our model reproduces plateau potentials in response to spatiotemporally clustered synaptic inputs to distal dendritic spines, and we then investigated the effect of both spatially clustered and spatially dispersed excitatory synaptic inputs on spine calcium elevation.

To first verify that plateau potentials occur in the model as in reported experiments, we simulated the model with synchronous synaptic input to 1–18 spines within an 18 μm dendritic segment located at increasing distances along a single terminal dendritic branch (*Figure 2A–B*). The model SPN exhibits a non-linear plateau potential when a cluster of distal, but not proximal, dendritic spines are simultaneously stimulated (we refer to stimulated synapses on dendritic spines as 'stimulated spines') (*Figure 2A*), consistent with experimental observations (*Du et al., 2017*; *Plotkin et al., 2011*). Blockade of NMDARs abolished the plateau potential, whereas blockade of VGCCs attenuated the plateau potential, consistent with previous experiments (*Plotkin et al., 2011*). Having confirmed that the electrical response is consistent with published results, we focused the remainder of our investigation on spine calcium concentration because of the importance of calcium for synaptic plasticity.

Supralinear spine calcium transients occur during simultaneous synaptic input to spatially clustered spines located on the distal portion of a dendritic branch. (*Figure 2B*). Stimulation at proximal locations shows a sublinear peak spine calcium response for up to 18 simultaneously stimulated spines. At distal locations, a sharp threshold emerges where the stimulation of a single additional spine produces a supralinear spine calcium elevation. Above threshold, stimulation of additional spines produces a graded increase in the magnitude of the spine calcium elevation. The threshold is distance-dependent, with fewer stimulated spines required to produce a supralinear calcium response closer to the terminal end of the dendritic branch. These results suggest that the distal dendritic spines of SPNs may be important sites for synaptic cooperativity—the ability of synapses to evoke supralinear responses when stimulated together. Furthermore, the supralinear spine calcium elevation evoked by clustered synaptic inputs may be an important mechanism for synaptic plasticity.

A critical question is whether spatial clustering of cortical inputs is required, or if spatially dispersed inputs can still cooperate to produce plateau potentials and supralinear spine calcium elevation. To address this question, excitatory inputs were randomly distributed over the proximal (27 – 119 μm from soma) or distal (135 – 225 μm from soma) regions of a dendritic branch, or over the entire branch (*Figure 2C*). Synaptic inputs that are spatially dispersed over the distal half of the branch still produce a supralinear response, although it is slightly smaller than the response to spatially clustered inputs. Too much spatial dispersion is not tolerated, as distributing 20 inputs over the entire branch no longer elicits supralinear spine calcium elevation. These results suggest that simultaneously stimulated synapses on distal dendritic spines still produce robust calcium influx when distributed within a 90 μm segment, indicating that close spatial clustering may not be a strong requirement for synaptic plasticity. In summary, the simulations of spatially dispersed synaptic inputs predict that SPNs produce plateau potentials and supralinear spine calcium elevation in response to the coordinated stimulation of ~16 excitatory synaptic inputs on a distal dendritic branch, and that this effect does *not* require precise spatial clustering.

## Spine calcium transients exhibit specificity for stimulated vs. non-stimulated spines

Synapse-specificity—that potentiation is limited to only those synapses which actively contribute to a postsynaptic response—is critical for synaptic plasticity to underlie learning and memory. Synapse-specific LTP requires that the elevated calcium concentration is confined to stimulated spines, as opposed to non-stimulated, neighboring spines, during a plateau potential. It is not clear how synapse-specificity is maintained during plateau potentials, when the entire dendritic branch becomes strongly depolarized and experiences calcium elevation as shown in calcium-imaging experiments (*Plotkin et al., 2011*). The strong dendritic depolarization could lead to calcium influx through VGCCs in neighboring non-stimulated spines or diffusion from dendrite to spine, leading to

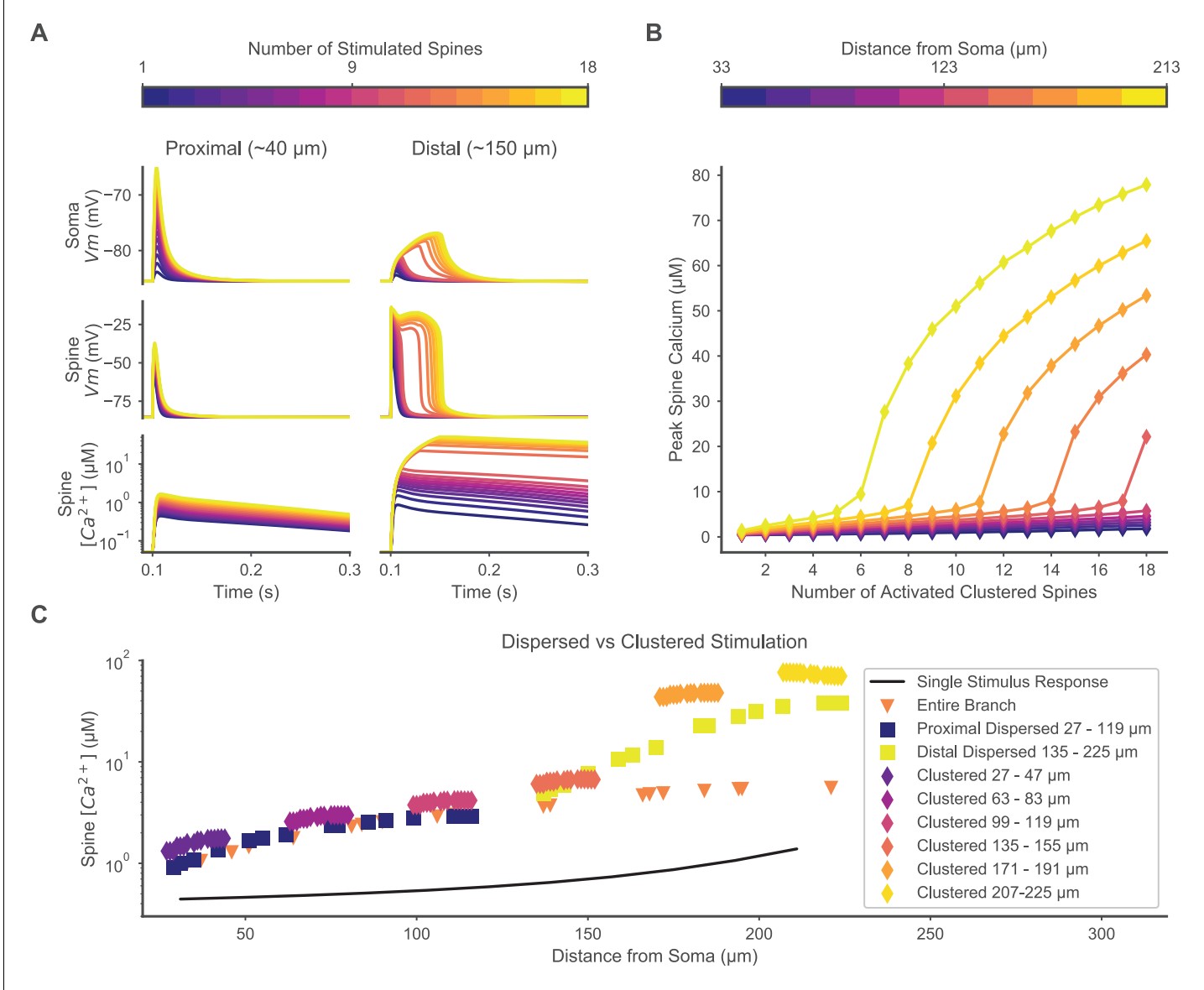

**Figure 2.** Clustered or dispersed distal synaptic inputs produce supralinear spine calcium elevation. (**A**) Synchronous stimulation of 1 – 18 clustered spines located distally (right) but not proximally (left) produces supralinear duration of depolarization at the soma (top) and stimulated spine (middle), and supralinear spine calcium elevation (bottom) for 12 or more spines. Spine membrane potential and calcium concentration traces are shown for only one spine per simulation. (**B**) Varying the location of 1 – 18 clustered synaptic inputs along the dendrite shows a distance-dependent threshold for supralinear spine calcium elevation, with fewer spines required at increasing distances from the soma. (**C**) Distal dispersed synaptic inputs also evoke supralinear calcium elevation. Peak spine calcium on a log scale is shown for each synaptically stimulated spine, for various spatial distributions of synaptic inputs, vs. the distance of each stimulated spine from the soma. Clustered diamonds indicate clustered stimulation of 16 spines with increasing distance from the soma, showing supralinear calcium distally. Spatially distributing 16 inputs distally (yellow squares) or proximally (dark blue squares) shows that distally located spatially dispersed synaptic inputs still produce supralinear spine calcium elevation. Distributing 20 inputs over the entire branch (orange triangles) fails to evoke supralinear calcium elevation. Synaptic inputs spatially dispersed over the proximal half of the branch reduce spine calcium elevation compared to the proximally clustered inputs, indicating that a small level of cooperativity can occur among proximal synaptic inputs despite the absence of a plateau potential. The unitary peak spine calcium for stimulation of only one spine at various distances from the soma is shown (black line) for reference.

DOI: https://doi.org/10.7554/eLife.38588.006

heterosynaptic LTP and a loss of synapse-specificity. To evaluate the extent of synapse-specificity, we investigated whether the calcium response differs between stimulated and neighboring non-stimulated spines for both spatially clustered and distributed inputs to a single dendritic branch (*Figure 3*).

To quantify synapse-specificity, we sampled calcium concentration from spines along the entire branch, both proximal and distal spines, including non-stimulated spines located between two stimulated spines for the spatially dispersed condition. In response to synaptic inputs to distal dendritic spines, peak calcium elevation in stimulated spines was an order of magnitude higher than in non-stimulated, neighboring spines. Similarly, when synaptic inputs were spatially dispersed, peak calcium elevation in stimulated spines was again an order of magnitude higher than in non-stimulated, interspersed spines (*Figure 3A*). Thus, calcium elevation may provide a synapse-specific signal for potentiation of coordinated synaptic inputs to spines on the distal dendritic branch, whether the synaptic inputs are spatially clustered or dispersed.

Although calcium elevation in non-stimulated spines was significantly smaller than stimulated spines, it was also significantly higher than baseline, although only for a subset of non-stimulated spines. Specifically, only the distally located non-stimulated spines exhibited a calcium elevation, with negligible calcium elevation in proximal non-stimulated spines. This suggests that heterosynaptic effects that may occur during a plateau potential would be limited to neighboring non-stimulated spines located on the distal dendritic branch. This distal-to-proximal gradient in calcium elevation is also observed in the dendritic shaft. Therefore, other calcium-dependent types of plasticity, such as homeostatic scaling or branch strength plasticity, also may occur in the distal dendritic shaft during plateau potentials. In summary, these results show three distinct, spatially specific calcium responses to coordinated distal synaptic inputs: a high, supralinear response in stimulated spines; an intermediate response in non-stimulated distal spines and in the distal dendritic shaft; and a negligible response in proximal non-stimulated spines.

## Mechanisms underlying spine-specific calcium dynamics

Understanding the biological mechanisms controlling differences in calcium dynamics between stimulated and neighboring non-stimulated spines may yield greater insights to synapse-specific signaling. Three different sources of spine calcium elevation—calcium permeable synaptic channels, voltage-gated calcium channels, and diffusion (of calcium or calcium-bound buffers)—could enhance or decrease differences in spine calcium concentration between stimulated and non-stimulated spines. To investigate which of these mechanisms distinguishes stimulated spines, we analyzed membrane potential, spine calcium concentration, and calcium channel currents in a stimulated spine and a neighboring non-stimulated spine during clustered distal synaptic stimulation. Additionally, we isolated VGCC-mediated and diffusion-mediated calcium elevations in non-stimulated spines by selectively blocking diffusion between non-stimulated spines and the dendritic shaft.

Calcium influx through synaptic channels underlies the high spine calcium elevation specific to stimulated spines. Analysis of spine head membrane potential indicated little difference between stimulated and neighboring non-stimulated spines (*Figure 3B*), indicating that VGCC-mediated calcium elevation would be similar in neighboring stimulated or non-stimulated spines. Analysis of VGCC calcium currents (*Figure 3D*) confirmed that VGCCs were activated similarly in stimulated and non-stimulated spines, whereas analysis of synaptic calcium currents in stimulated spines indicated orders of magnitude higher currents than VGCC currents in stimulated spines (*Figure 3E*).

VGCCs and diffusion both contribute to calcium elevations in non-stimulated neighboring spines. Blocking diffusion of calcium and calcium buffers between the dendritic shaft and non-stimulated spines reduced the later phase of calcium elevation in non-stimulated spines (*Figure 3C*), indicating that diffusion may support calcium transient duration. In contrast, the early phase and peak amplitude of the calcium transient did not change when diffusion between the non-stimulated spines and dendritic shaft was blocked, indicating that VGCCs on non-stimulated spines underlie calcium transient peak amplitude.

Together, these results indicate that the spatial specificity of calcium elevation results from synaptic calcium influx in stimulated spines, whereas both diffusion and VGCC influx increase calcium concentration in neighboring, non-stimulated spines. Consequently, calcium transients exhibit a robust synapse-specific signal in stimulated spines despite highly similar membrane potentials in neighboring spines during plateau potentials. Further, as membrane potential is sharply attenuated as

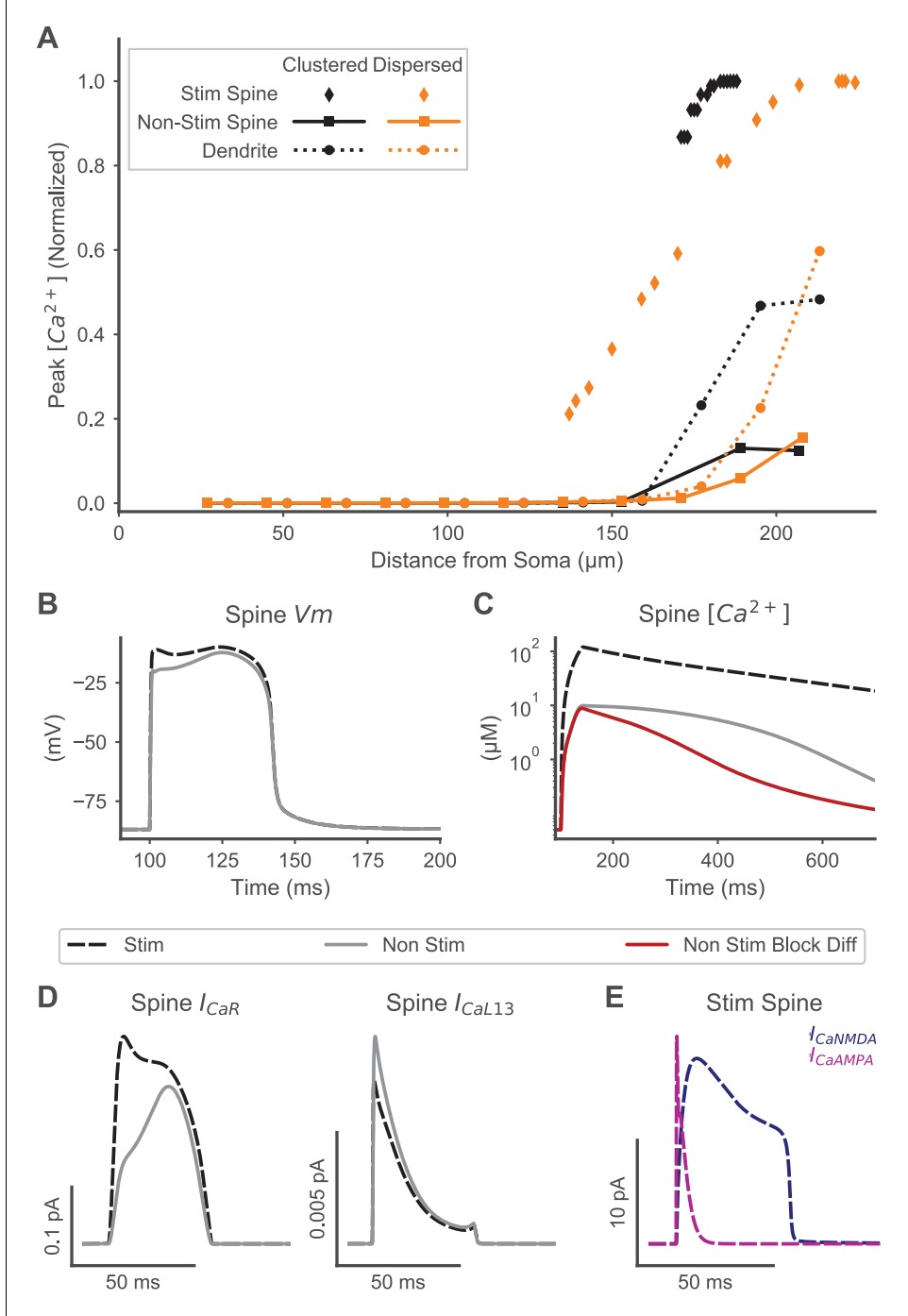

**Figure 3.** Spine calcium elevation exhibits spatial specificity for both clustered and distributed synaptic inputs. (A) Supralinear calcium elevation is limited to stimulated spines (diamonds) with neighboring non-stimulated spines (squares) exhibiting an order of magnitude lower peak calcium for both clustered (orange) and dispersed (black) inputs. Calcium elevation in the dendrite (circles) and non-stimulated spines also exhibits spatial specificity, with smaller calcium elevations limited to the distal dendrite near the site of stimulation and negligible calcium elevations observed proximally. (B) Spine membrane potential (Vm) is similar for a stimulated (black dashed) or non-stimulated neighboring (gray line) spine during clustered stimulation (as in (A)). (C) Spine calcium concentration (corresponding to Vm in (B)) is an order of magnitude higher in a stimulated spine than a non-stimulated neighboring spine. When diffusion is blocked (red trace) between non-stimulated spines and the dendritic shaft, VGCC contributions to non-stimulated spine calcium elevation are isolated, revealing that VGCCs contribute to the peak calcium, but diffusion contributes to the later phase of calcium elevation in non-stimulated

*Figure 3 continued on next page*

*Figure 3 continued*

neighboring spines. (**D**) VGCC currents in a stimulated and neighboring non-stimulated spine (as in (**B-C**)) are shown for a high voltage activated (CaR, left) and low voltage activated (CaL1.3, right) channel, indicating that VGCCs contribute to calcium elevation in both spines. Note that (**B-D**) share a common legend. (**E**) Synaptic calcium currents are shown for the stimulated spine, indicating that NMDAR and Ca-permeable AMPAR channels underlie the stimulus-specific spine calcium elevation.

DOI: https://doi.org/10.7554/eLife.38588.007

depolarization propagates towards the soma, the VGCC-mediated elevation is limited to distal non-stimulated spines whereas proximal non-stimulated spines exhibit negligible calcium elevation. The distinct calcium transients and sources observed may therefore support stimulus-specific synaptic plasticity in stimulated spines and heterosynaptic plasticity in neighboring, non-stimulated spines.

## Synaptic cooperativity is strongest for synaptic inputs located on the same dendritic branch

Synaptic inputs are likely to be spatially and temporally dispersed on multiple dendritic branches in vivo. This raises critical questions: to evoke supralinear spine calcium transients, do cortical inputs need to target a single dendritic branch, or can inputs be spatially dispersed on the entire neuron? Further, would synaptic inputs on multiple branches act independently—such that the number of synaptic inputs per branch required to evoke supralinear calcium transients is independent of synaptic inputs to other branches—or would they interact (cooperate) to lower the threshold number of synaptic inputs per branch required for supralinear calcium transients? To address these questions, we investigated the effect of spatial dispersion of synaptic inputs to multiple dendritic branches. Synaptic inputs were randomly distributed on two tertiary branches with a common secondary branch parent; on four tertiary branches with a common primary branch parent; or on eight tertiary dendritic branches. Additionally, we used both simultaneous synaptic inputs and temporal dispersion, created from random, exponentially distributed intervals. Simulations used average interstimulus intervals (ISIs) of 2.5, 5, or 10 ms between synaptic inputs on each branch (e.g. with a 10 ms mean ISI per branch, when two branches are stimulated, the overall mean ISI is 5 ms for all inputs) (*Figure 4*). Specifying the ISI per branch has the advantage that the total duration of synaptic stimulation is independent of the number of branches. The temporal order of stimulated spines was randomly selected and followed no spatial pattern.

When simultaneous synaptic inputs are spatially dispersed across multiple branches, the total number of synaptic inputs required to elicit supralinear spine calcium transients is increased (*Figure 4B*). This indicates a degree of dendritic branch independence. However, slightly fewer stimulated spines per branch are required to evoke supralinear spine calcium transients when more branches are stimulated, indicating a small degree of interaction between branches. Thus, supralinear calcium elevation in stimulated spines is dendritic branch-specific, although the required number of stimulated spines on a single branch may be slightly reduced by interaction with synaptic inputs to other dendritic branches.

As the average ISI is increased, there is an overall reduction in supralinear spine calcium elevation. However, when multiple branches are stimulated, increasing the mean ISI up to 5 ms can increase spine calcium elevation, indicating that, with spatial dispersion, a small temporal dispersion increases cooperativity. The corresponding somatic voltage (*Figure 4C*) shows sustained depolarization, which may be causing the increased spine calcium elevation. Supralinearity in spine calcium elevation appears to be negligible for average ISIs greater than 10 ms. This suggests that there is a limited temporal window for evoking supralinear spine calcium transients in SPNs, although precisely synchronous inputs are not required.

Altogether, results using spatiotemporally dispersed synaptic inputs suggest that each distal dendritic branch in SPNs may function as a relatively independent subunit for integrating synaptic inputs with spine-specific calcium responses. These results are consistent with the theory that individual dendritic branches serve as critical subunits for synaptic integration and plasticity (*Branco and Häusser, 2010*).

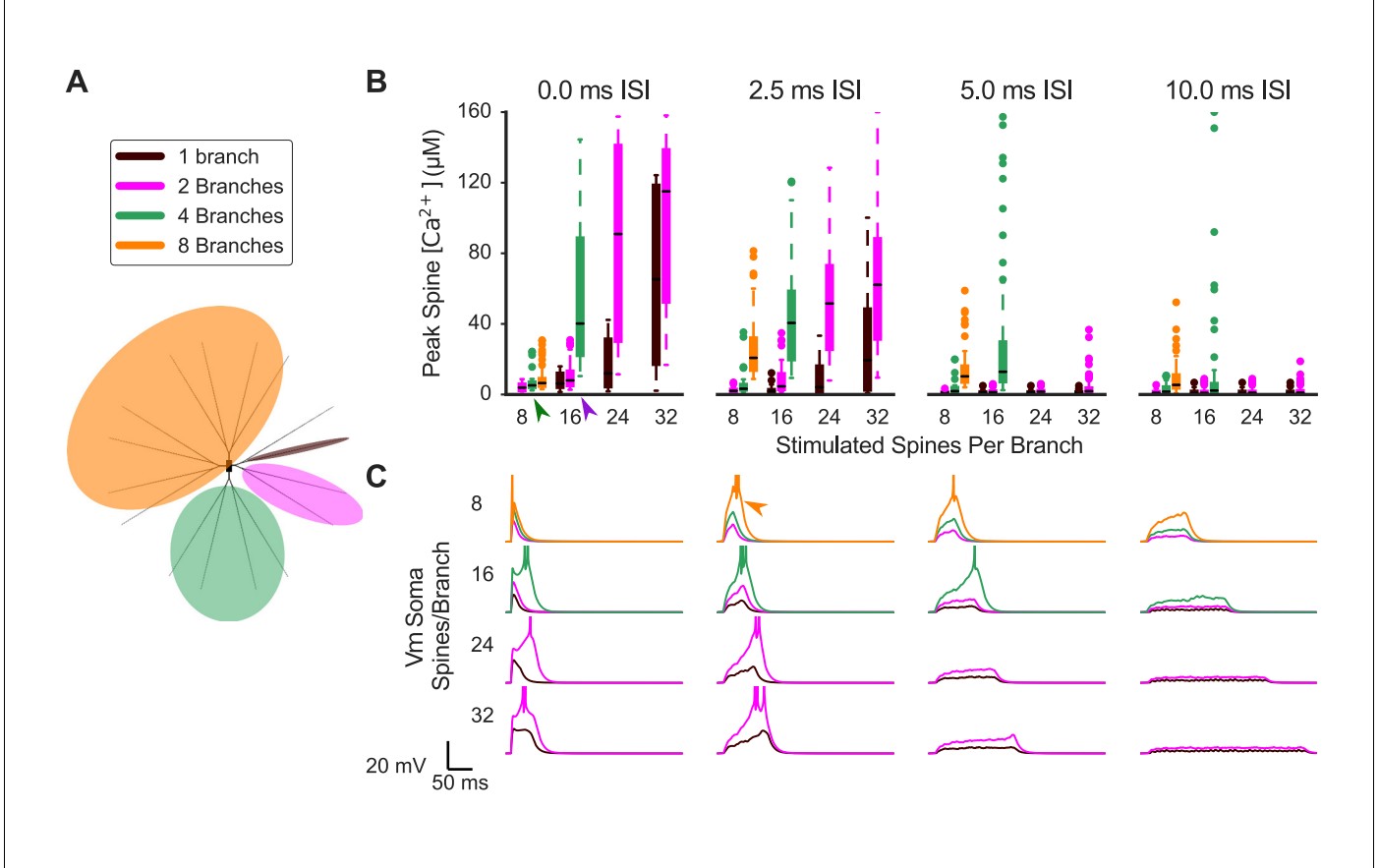

**Figure 4.** Synaptic cooperativity is strongest for synaptic inputs located on the same dendritic branch. (A) Schematic indicates spatiotemporally distributed inputs to one, two, four, or eight dendritic branches (legend colors also correspond to (B) and (C). (B) Peak spine calcium for all stimulated spines is shown as a box and whisker plot with outliers for various patterns of spatiotemporally distributed inputs. Eight, 16, 24, or 32 spines per branch are stimulated with one, two, four, or eight total branches being stimulated for 0, 2.5, 5, or 10 ms mean interstimulus interval per branch (panels left to right). Stimulating multiple branches with the same number of spines/branch does not strongly affect spine calcium elevation (e.g. green arrowhead, eight spines per branch on two to eight branches with 0.0 ISI), indicating branch independence. However, near threshold (e.g. purple arrowhead, 16 spines/branch on two versus four branches for 0.0 ms ISI), stimulating multiple branches enhances spine calcium elevation, indicating small branch interactions whereby inputs to other branches can reduce the threshold of required inputs on a branch for supralinear spine calcium elevation. Spine calcium elevation decreases with increasing ISI (i.e. decreased peak calcium in panels from left to right), indicating that cooperation among synaptic inputs is limited to a temporal window of 10 s of ms. Note that for temporally distributed simulations, the ISIs are exponentially distributed with the average value indicated. The average ISI per branch means (for example) that an average ISI of 10 ms per branch corresponds to a total average ISI of 5 ms for all inputs when two branches are stimulated, and 2.5 ms for all inputs when four total branches are stimulated. The order of inputs was randomly applied and followed no spatial pattern. (C) Soma membrane potential corresponding to spine calcium concentration measures in (B). Plateau potential duration corresponds to spine calcium supralinearity. Action potentials are clipped in voltage traces. In some cases (e.g. orange arrowhead), increasing ISI from 0 to 2.5 ms enhances cooperativity among synaptic inputs, as it prolongs the duration of the plateau potential.

DOI: https://doi.org/10.7554/eLife.38588.008

## Distally evoked plateau potential facilitates calcium influx in proximal stimulated spines

The distinct calcium signals in proximal versus distal stimulated spines in response to coordinated synaptic inputs suggests that proximal and distal synapses may have distinct functions in SPNs. This raises the question of whether distal and proximal synapses can interact or instead function independently. Just as a somatic action potential can back-propagate to enhance spine calcium elevation, we investigated whether a cluster of synaptic inputs, in the absence of a somatic action potential, can enhance spine calcium elevation. To test whether a distally evoked plateau potential could interact with a stimulated proximal synapse on the same branch, we paired clustered distal stimulation

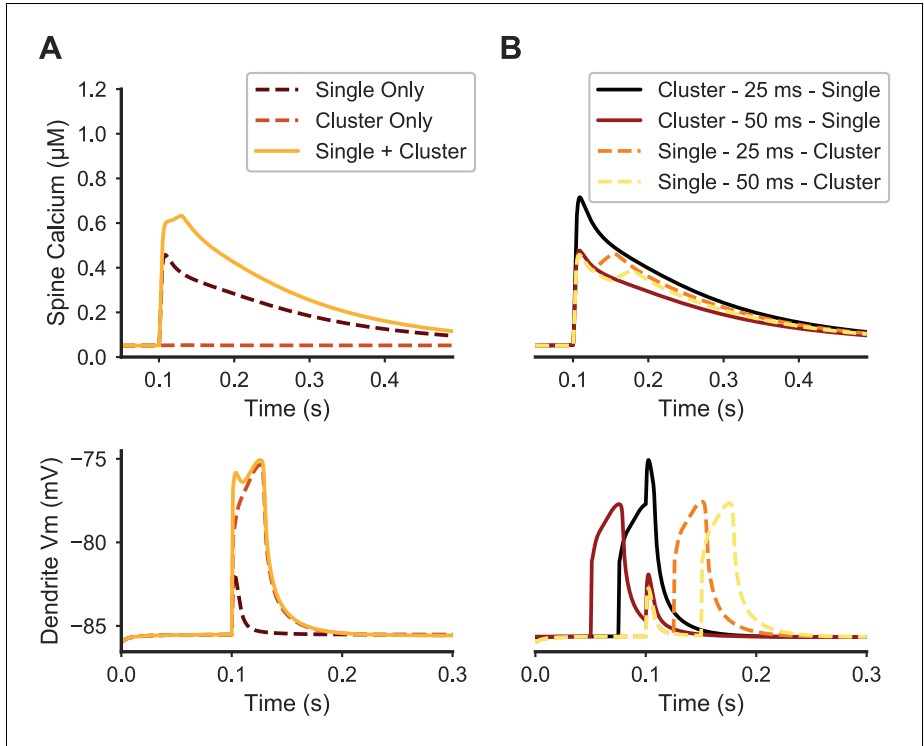

**Figure 5.** Paired stimulation of clustered distal spines and a single proximal spine produces elevated calcium in the proximal spine. (**A**) Spine calcium traces (top) are shown for a single proximal spine, either when synaptically stimulated alone (dark dashed line); unstimulated during synaptic stimulation of clustered distal spines (light dashed line); or stimulated simultaneously with clustered distal stimulation (solid yellow line). Proximal spine calcium elevation is higher when stimulated with clustered distal stimulation than the sum of the proximal spine calcium responses to only distal cluster stimulation and only proximal spine stimulation. Corresponding membrane potential at the dendritic shaft is shown below. (**B**) Temporal dependence governing the interaction between proximal synaptic stimulation (same y axis as (**A**)) and synaptic stimulation of a distal cluster of spines. Traces are temporally aligned to the onset of stimulation of the proximal spine synapse, while paired with distal clustered spine stimulation at varying temporal intervals. When the distal cluster stimulation precedes the single proximal synaptic input by 25 ms (black trace), elevated calcium is produced in the proximal spine, but not for a longer delay (50 ms) or when the single proximal stimulus precedes the distal clustered stimulation. Dendritic membrane potential traces (bottom) show the temporal dependence of peak calcium facilitation on depolarization in the proximal dendritic shaft; in each trace, the proximal stimulus occurs at 0.1 ms on the x axis.

DOI: https://doi.org/10.7554/eLife.38588.009

and single proximal spine stimulation with varying temporal intervals and evaluated the calcium responses in the proximal spine (*Figure 5*).

Our results show that distal clustered synaptic inputs can facilitate calcium elevation in response to synaptic input to proximal spines, dependent on timing. Simultaneous stimulation of a distal cluster and single proximal spine facilitated the proximal spine peak calcium concentration (*Figure 5A*). The simultaneous stimulation produced higher proximal spine calcium elevation than the sum of the proximal and distal stimulations performed independently. We repeated the pairing at four different ISIs—proximal stimulus 25 or 50 ms before distal cluster stimulus, and proximal stimulus 25 or 50 ms after the distal cluster (*Figure 5B*). We found that peak calcium elevation in the proximal spine was facilitated if the proximal stimulus came 25 ms after the distal stimuli. However, peak calcium elevation in the proximal spine was not facilitated when the proximal spine stimulus came 50 or 25 ms before the distal stimuli, nor when the proximal stimulus came 50 ms after the distal stimuli. Interestingly, the duration of the calcium transient in the proximal spine was prolonged when the proximal spine was stimulated prior to the distal cluster. To investigate the mechanism underlying this temporal dependence, we evaluated the voltage in the dendritic shaft at the base of the proximal spine. The enhancement of peak calcium in the proximal spine correlated with the amplitude of the

depolarization propagating from the distally evoked plateau potential. Together, these results indicate that distally evoked plateau potentials can interact with proximal synaptic inputs within a 25 ms temporal window, and that the interaction shows asymmetrical dependence on timing. Thus, similar to spike-timing-dependent plasticity rules which depend on the order and timing of presynaptic activation and a postsynaptic spike, we predict that a plateau potential originating in the distal dendrite may facilitate plasticity when paired with stimulation of proximal synapses.

## Inhibition attenuates calcium elevation in stimulated spines

Multiple sources of inhibitory, GABAergic synapses on SPNs may regulate synaptic integration and spine calcium transients. The ionotropic GABA$_A$ synapses exhibit distinct spatial organization, with fast spiking interneurons (FSIs) targeting proximal dendrites and low threshold spiking interneurons (LTSIs) and SPN collaterals targeting distal dendrites, and they exhibit distinct temporal kinetics, with neurogliaform (NGF) interneurons exhibiting much slower activation and inactivation time constants (*Ibáñez-Sandoval et al., 2011*; *Straub et al., 2016*; *Tepper et al., 2010*). To make predictions about synaptic integration and plasticity that may be relevant in vivo, it is critical to include the effects of inhibition in our investigation, as inhibitory signaling is extensive in the striatum. Thus, we investigated the effects of GABA$_A$ kinetics, location, and timing relative to clustered glutamatergic input on regulating plateau potentials and spine calcium elevation (*Figure 6*).

Our results demonstrate that non-linear spine calcium dynamics depend strongly on the location of simultaneously stimulated GABA$_A$ synapses (*Figure 6B*). We paired stimulation of a single GABAergic synapse on the dendritic shaft with simultaneous stimulation of excitatory synapses on a cluster of distal dendritic spines, and repeated simulations while varying the location of the single GABAergic synapse. We ran the same set of simulations for both the slow and fast GABA$_A$ kinetics. Distal GABAergic inputs near the site of clustered excitatory synaptic inputs have the strongest inhibitory effect on spine calcium elevation, reducing the response by ~50% for slow GABA$_A$ kinetics, whereas proximally located GABAergic synapses have little effect on spine calcium. The effect was similarly distance-dependent but weaker for fast GABA$_A$ kinetics, with a 25% reduction in spine calcium elevation. These results suggest that distally located GABAergic synapses from LTSIs, SPN collaterals, or NGF interneurons likely regulate the occurrence of supralinear spine calcium influx in response to clustered glutamatergic stimulation, whereas FSIs with proximal synapses do not regulate distal spine calcium dynamics.

Additionally, our results show that the timing of GABAergic stimulation relative to clustered glutamatergic stimulation affects the resulting spine calcium elevation (*Figure 6A*). We paired stimulation of a single GABAergic synapse located on the distal dendritic shaft with stimulation of excitatory synapses on a cluster of distal dendritic spines at the same dendritic location, and repeated simulations varying the timing between GABAergic and glutamatergic stimulation. Again, we ran the same simulations for GABAergic synapses with either slow or fast GABA$_A$ kinetics. For both slow and fast GABA$_A$ kinetics, the timing of inhibition relative to excitation strongly affected spine calcium elevation. For the slow GABA$_A$ kinetics, GABAergic input from 100 ms before to 25 ms after clustered glutamatergic stimulation diminished spine calcium elevation, whereas the fast GABA$_A$ kinetics have a much narrower temporal window for inhibiting the calcium response. Together, these results indicate that inhibition may strongly regulate spine calcium dynamics when active prior to or concurrent with clustered glutamatergic input. In particular, GABA$_A$ synapses with slow kinetics (i.e. from NGF interneurons) may be particularly potent regulators of dendritic integration and synaptic interactions in SPNs.

As FSIs provide strong proximal inhibition to SPNs in vivo, we also investigated whether a train of proximal GABAergic inputs (as opposed to a single input) would affect cooperativity among stimulated spines or between dendritic branches (*Figure 6C*). A train of 20 GABAergic inputs (ISI = 3 ms) was applied to the proximal, primary dendritic branch while 16 or 32 glutamatergic inputs per branch were dispersed over the entire branch on one or two neighboring tertiary branches with an ISI of 2.5 ms per branch. The onset of the FSI input train and the glutamatergic stimulation was simultaneous. Interestingly, we found that a train of proximal GABAergic stimulation enhanced spine calcium responses when glutamatergic inputs were sub- or near-threshold (16 spines/branch on one or two branches), but not when above threshold (32 spines/branch on two branches) for supralinear spine calcium elevation. We assessed whether the effect of FSI input trains on spine calcium elevation depended on distance of stimulated spines from the soma (*Figure 6—figure supplement 1*),

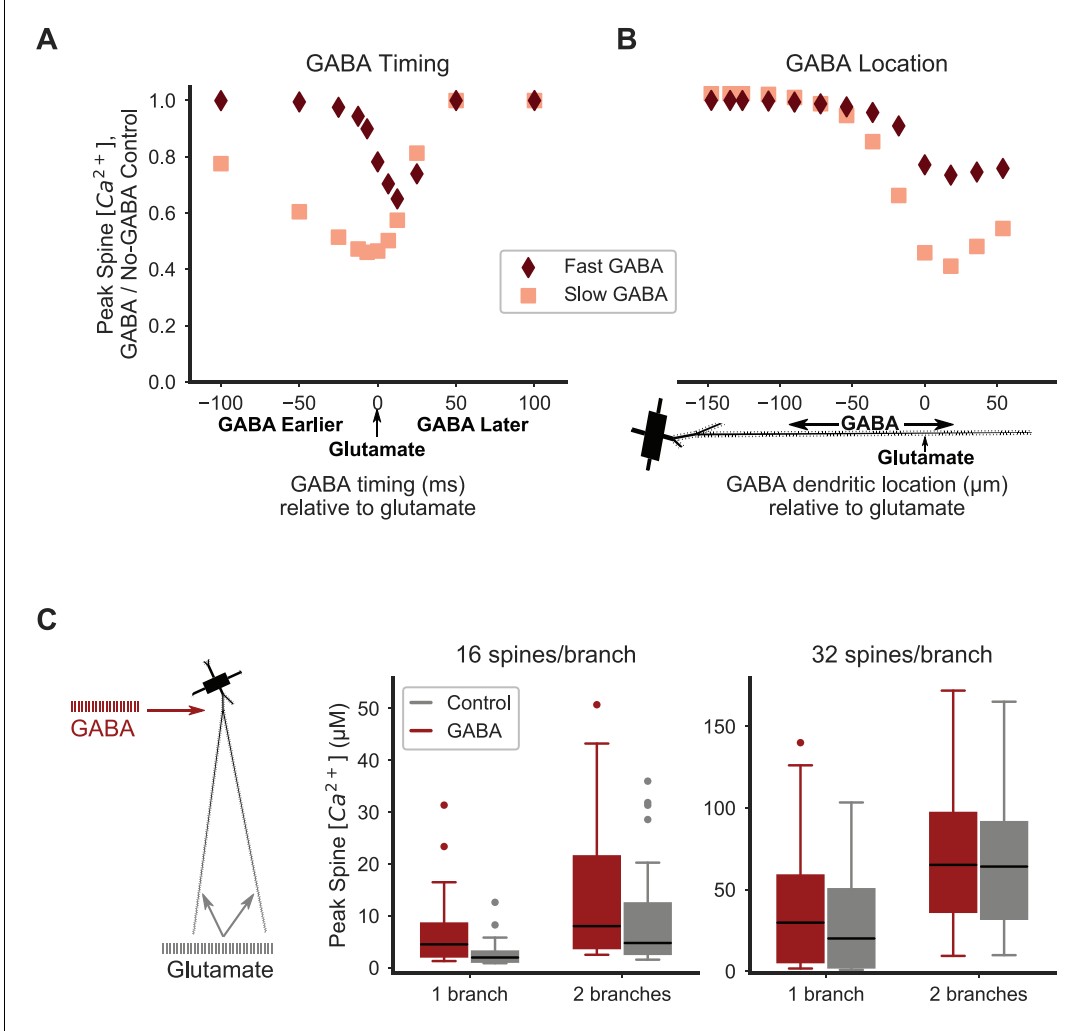

**Figure 6.** Distal inhibition attenuates synaptic cooperativity while proximal inhibition enhances synaptic cooperativity. (A) Attenuation of spine calcium elevation by inhibition is timing-dependent. GABAergic stimulation attenuates the spine calcium elevation evoked by coordinated glutamatergic stimulation when GABAergic stimulation occurs between 25 ms before to 50 ms after glutamateric stimulation for fast $GABA_A$ kinetics, or 100 ms before to 50 ms after glutamatergic stimulation for slow $GABA_A$ kinetics. The GABAergic synapse was located on the dendritic shaft at the same location as clustered glutamatergic stimulation of 16 distally located neighboring spines. (B) Attenuation of spine calcium elevation by inhibition is location-dependent. GABAergic stimulation simultaneous with glutamatergic stimulation (of 16 distal, neighboring spines) attenuates supralinear spine calcium elevation when the GABAergic stimulation is located distally, near the clustered glutamatergic stimulation, whereas single proximal GABAergic synaptic inputs have little effect. For both A-B, peak spine calcium is normalized to the no-GABA control, and fast (diamonds) or slow (squares) $GABA_A$ kinetics correspond to $GABA_A$ synapses from SPNs, LTSIs, FSIs (fast), or NGFs (slow). (C) Strong proximal inhibitory input, corresponding to FSIs, enhances supralinear spine calcium elevation for sub- or near-threshold stimulations (16 spines per branch on one or two branches). The peak calcium elevation in all stimulated spines is shown as box-and-whisker plots. The effect of inhibition (GABA; red bars) on peak spine calcium in stimulated spines is higher than control (gray bars) for 16 spines per branch (left) but not 32 spines per branch (right). Both excitatory and inhibitory synapses were stimulated during the same time frame in these simulations. *Figure 6—figure supplement 1* shows the peak spine calcium vs. dendritic location for each stimulated spine.

DOI: https://doi.org/10.7554/eLife.38588.010

The following figure supplement is available for figure 6:

**Figure supplement 1.** Distal spines exhibit elevated calcium in the presence of strong proximal inhibition.

DOI: https://doi.org/10.7554/eLife.38588.011

and we found that distal spines, but not proximal spines, exhibit elevated calcium in response to proximal FSI inputs relative to the control (no FSI) condition. Our finding that proximal GABAergic stimulation may enhance spine calcium elevation is consistent with experiments demonstrating that when SPNs are in the hyperpolarized downstate, GABAergic stimulation produces depolarization (*Blackwell et al., 2003*; *Bracci and Panzeri, 2006*). This depolarization may propagate to distal dendritic spines to enhance supralinear spine calcium elevation in stimulated spines when SPNs are in a hyperpolarized downstate.

## Inhibition enhances stimulus-specificity of spine calcium transients

Notably, although we observed above that inhibition close to excitatory inputs can reduce the magnitude of spine calcium elevation in stimulated spines (likely by lowering dendritic propagation of potentials), inhibition neither abolished the plateau potential nor fully blocked supralinear spine calcium elevation. This raised the intriguing possibility that the reduced local membrane resistance caused by inhibition may also influence non-stimulated spines. Thus, we evaluated the effect of inhibition by measuring the peak calcium response in non-stimulated spines relative to the peak calcium response in stimulated spines in the presence or absence of a co-located and simultaneously stimulated GABAergic synapse during simultaneous excitatory stimulation of a distal cluster of dendritic spines.

The most significant functional effect of distal GABAergic synaptic input is to enhance spatial specificity (*Figure 7*). When compared to the control (no-GABAergic input) condition, stimulation of a single GABAergic synapse on the distal dendrite reduces the ratio of peak calcium elevation in non-stimulated spines relative to the peak in stimulated spines (*Figure 7A*). Further, GABAergic stimulation narrows the spatial extent of calcium influx in non-stimulated spines and in the dendritic shaft (*Figure 7B*), reducing the ratio of non-stimulated spine peak calcium to stimulated spine peak calcium much more for distant than adjacent spines and thereby limiting the spatial extent of heterosynaptic calcium elevation. As calcium elevation can lead to both LTP and LTD, with lower levels of calcium elevation associated with LTD and higher levels associated with LTP, inhibition may critically regulate heterosynaptic plasticity by reducing calcium elevation in non-stimulated spines from a level that could lead to LTP to a level that could lead to LTD, thereby preserving synapse-specific potentiation.

It is critical in computational modeling to assess the robustness of results to variations in parameter values. To assess the robustness of the effect of inhibition on synapse-specificity, we systematically varied conductances of voltage-gated and synaptic channels, calcium buffer quantities and pump densities, and spine neck axial resistance by ±10% and 20%. For each condition, we computed the specificity ratio as the ratio of peak calcium of non-stimulated spines to peak calcium of stimulated spines, such that a smaller value indicates higher specificity. We then divided the specificity ratio observed with inhibition by the specificity ratio observed without inhibition (called GABA/No GABA specificity ratio), such that a value <1 indicates that inhibition enhances specificity, while a value >1 indicates that inhibition reduces specificity. Inhibition consistently enhanced spatial specificity across all parameter variations, except for a 20% decrease in NMDAR conductance (*Figure 7— figure supplement 1*). The magnitude of the effect of inhibition was most sensitive to NMDAR, CaR, and GABAR conductances. As expected, a larger GABAR conductance enhanced the GABA/No GABA specificity ratio. Larger inward currents reduced specificity, and the large effect with NMDAR and CaR conductances confirms our previous results showing that these are critical parameters for the supralinear spine calcium response. Surprisingly, decreases in NMDAR or CaR conductances also reduced the GABA/No GABA specificity ratio. When either CaR or NMDAR conductance is lowered, stimulation is below threshold, and the reduced activation of VGCCs greatly enhances spatial specificity in the absence of inhibition. The sensitivity to CaR and NMDAR conductances suggests that neurons may regulate the balance of these channels; thus, we repeated simulations with an increase of CaR and a decrease of NMDAR (and vice versa, no attempt was made to balance these changes). As predicted, sensitivity to the paired parameter change was smaller than sensitivity to a single parameter change. In summary, this parameter sensitivity analysis suggests that the balance of NMDAR and CaR channels may fine-tune spatial specificity of spine calcium transients.

In addition to robustness to parameter variations, it is important that our main result does not depend on underlying assumptions in the modeling method. Specifically, our model of calcium dynamics assumes that deterministic, one-dimensional diffusion and reaction equations are sufficient

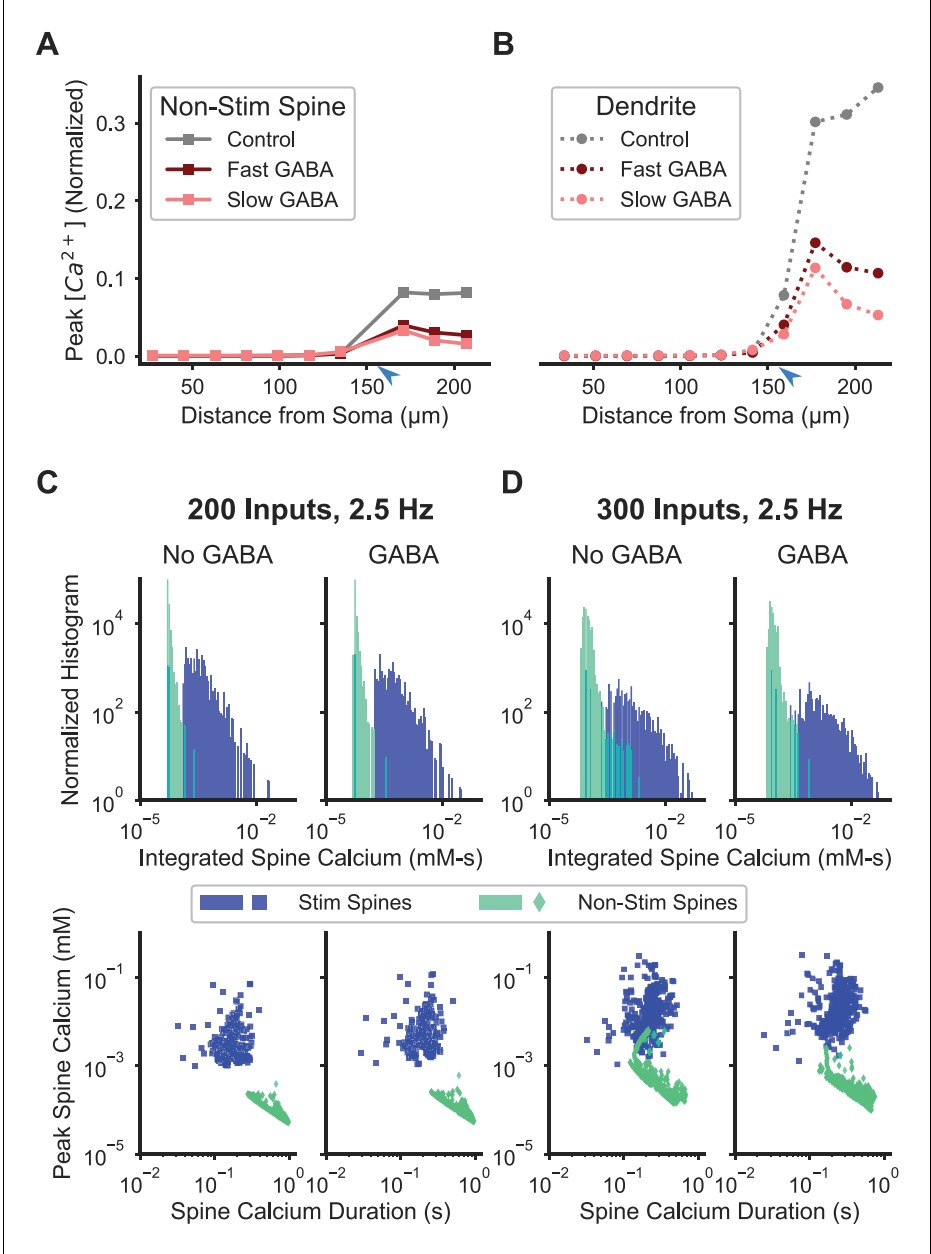

**Figure 7.** Inhibition enhances spatial specificity of spine calcium elevation. (**A–B**) Stimulation of a single GABAergic synapse on the distal dendrite reduces the ratio of peak calcium in nonstimulated spines (**A**) and dendritic shaft (**B**) relative to the peak in synaptically stimulated spines for fast or slow GABA$_A$ kinetics. For both fast and slow GABA$_A$ kinetics, the GABAergic synapse is on the dendritic shaft, co-located with and synchronously stimulated with glutamatergic stimulation of a cluster of 16 distal dendritic spines (location of synaptic input indicated by blue arrowhead). *Figure 7—figure supplement 1* shows the sensitivity of the effect of inhibition on spatial specificity to parameter variations. *Figure 7—figure supplement 2* shows the robustness of this main finding to model assumptions. (**C–D**) Inhibition enhances synapse-specificity during stimulation of randomly distributed excitatory synaptic inputs. Distributions of integrated spine calcium concentration for stimulated (blue) and non-stimulated (green) spines are shown for randomly distributed excitatory Poisson input trains (2.5 Hz) for lower (C; n = 200) or higher (D; n = 300) numbers of independent excitatory synaptic inputs, either with (right columns) or without (left columns) 50 GABAergic Poisson input trains (2.5 Hz). Top row shows the log-scale distribution of integrated spine calcium elevation over the 1 s duration inputs; bottom row shows log-scale scatter plots of peak calcium (y axis) versus normalized calcium duration (i.e. integrated calcium/peak calcium) (x axis).
DOI: https://doi.org/10.7554/eLife.38588.012

*Figure 7 continued on next page*

*Figure 7 continued*

The following figure supplements are available for figure 7:

**Figure supplement 1.** Parameter sensitivity analysis for enhancement of spatial specificity by inhibition.
DOI: https://doi.org/10.7554/eLife.38588.013

**Figure supplement 2.** Robustness of model assumptions for enhancement of spatial specificity by inhibition.
DOI: https://doi.org/10.7554/eLife.38588.014

to capture the dynamics of calcium elevation in dendritic spines. We evaluated whether the following variations affected the central finding that inhibition enhances synapse-specificity: (1) spatial and temporal discretization step size; (2) variations in calcium diffusion rates; (3) facilitated calcium diffusion between spine neck and dendritic shaft; and (4) variations in calcium influx and efflux caused by channels and pumps. Importantly, these findings did not qualitatively change the main result that inhibition enhances spatial specificity (*Figure 7—figure supplement 2*).

A critical functional question is whether inhibition regulates synaptic specificity for spatiotemporally distributed inputs, which are likely to occur in vivo. To answer this question, we applied both excitatory (n = 200 or 300) and inhibitory (n = 50) synaptic inputs with random spatial dispersion, with each individual synaptic input consisting of an independent Poisson process with an average frequency of 2.5 Hz. We assessed spine calcium with histograms of integrated calcium for the 1 s simulation duration, as well as scatter plots of peak values versus normalized duration (i.e. integrated calcium concentration/peak calcium concentration) for each spine. At the lower (n = 200) activity level, synaptic specificity was robust with or without inhibition, with little overlap in integrated calcium and no overlap for peak versus duration scatter points (*Figure 7C*). However, with higher (n = 300) activity and no inhibition, synaptic specificity was reduced, as overlap in both the histogram and scatter points occurred for stimulated and non-stimulated spines. Notably, when inhibition was included during higher activity, synaptic specificity was increased, indicating that inhibition may function in vivo to maintain spatial specificity of spine calcium transients (*Figure 7D*).

To quantify the extent to which inhibition increased separation between stimulated and non-stimulated spine calcium responses, we performed a cluster analysis using duration and peak calcium as parameters. For the cluster analysis, we performed K-means clustering on the unlabeled data, and then computed the confusion matrix—the number of correctly and incorrectly clustered data points in the unlabeled cluster analysis compared to the real, labeled dataset, where the label is whether the spine received synaptic input. We evaluated the number of incorrectly clustered spines in the confusion matrix and computed the distance between clusters as a metric for the effect of inhibition on synaptic specificity. For the higher activity level without inhibition, the cluster analysis incorrectly identified the stimulation status of 58 spines, whereas with inhibition, the stimulation status of only 19 spines was incorrectly identified (out of 3280 total spines). For the lower activity level, no spines were incorrectly identified, but inhibition increased the distance between cluster centroids by 5.3%. Altogether, these results suggest that although inhibition reduces calcium elevation in stimulated spines, its most significant effect is to enhance stimulus-specific calcium signaling and it may therefore regulate heterosynaptic plasticity.

## Discussion

We demonstrated that spatial and temporal patterns of synaptic input to SPNs produce spatially specific and stimulus-specific spine calcium responses in a biologically detailed computational model with sophisticated calcium dynamics. A sufficient number of excitatory inputs to a distal dendritic branch can cooperate to evoke supralinear spine calcium responses in the synaptically stimulated spines, whether the stimulated spines are clustered or dispersed along the distal portion of the branch. The resulting spine calcium responses exhibit specificity for synaptically stimulated spines, with an order of magnitude higher calcium influx than non-stimulated neighboring spines, and this spatial specificity is further enhanced by nearby GABAergic synaptic stimulation. Synaptic cooperativity exhibits dendritic-branch specificity, as synaptic inputs to one branch produce higher spine calcium responses than the same number of inputs distributed to multiple branches. Additionally, the calcium elevation in a proximal spine can be facilitated when the synaptic input to that spine occurs synchronous with or shortly following distally evoked plateau potentials, indicating potential

interactions between distal and proximal synaptic input. Together, our results suggest that spine calcium elevation is a stimulus-specific signal which can differentiate between synaptically stimulated and non-stimulated spines. We predict that these distinct calcium responses will enable both input-specific LTP and heterosynaptic LTD, and that inhibition critically regulates heterosynaptic activity.

In key ways, our results extend other experimental and modeling studies which have found that striatal spiny projection neurons exhibit plateau potentials in response to clustered excitation of neighboring spines on a distal dendritic branch (*Du et al., 2017*; *Plotkin et al., 2011*). First, our focus is on spine calcium dynamics, whereas previous studies focused on somatic voltage. *Plotkin et al. (2011)* did find robust calcium elevation at the site of clustered synaptic stimulation; however, our results additionally show the spatial specificity of spine calcium elevation. Second, our model predicts that spatial clustering within the branch is not required for supralinear spine calcium elevation or plateau potentials, if enough inputs are located distally on the same dendritic branch. SPNs receive few (one to three) synapses from any individual cortical neuron and convergent inputs from thousands of cortical neurons (*Kincaid et al., 1998*; *Zheng and Wilson, 2002*), and the likelihood of clustered, synchronous inputs to occur in vivo is unknown, so the ability for dispersed inputs to a distal branch to also evoke supralinear responses may increase the likelihood of supralinear spine calcium transients occurring in vivo.

Our finding that coordinated synaptic inputs produce supralinear spine calcium elevation suggests a mechanism for 'cooperative LTP' (LTP produced by stimulation of multiple synaptic inputs in the absence of a postsynaptic action potential) in striatal SPNs. Although the induction of cooperative LTP by clustered synaptic input has not been demonstrated in SPNs, it has been observed in pyramidal neurons in cortex and hippocampus (*Brandalise et al., 2016*; *Golding et al., 2002*; *Gordon et al., 2006*; *Harnett et al., 2012*; *Larkum et al., 2009*; *Losonczy et al., 2008*; *Makara and Magee, 2013*; *Schiller et al., 2000*; *Weber et al., 2016*). In pyramidal neurons, cooperative LTP requires NMDAR activation and can be induced by synaptic stimulation of a small number of neighboring spines on a thin dendritic branch. Similarly, we observe that supralinear spine calcium responses in SPNs are NMDAR-dependent and are achieved by coordinated synaptic inputs to a (thin) distal branch. The dependence of supralinear spine calcium elevation on spatially and temporally distributed inputs also is similar to observations in pyramidal neurons. Spatially and temporally distributed inputs to a single distal dendritic branch in CA3 pyramidal neurons have been shown to evoke NMDA spikes, for inputs spatially distributed within a 60 micron length of dendritic branch or temporally distributed with intervals up to 2 ms (*Makara and Magee, 2013*). These results are consistent with our findings that inputs spatially dispersed over the distal half of a dendritic branch or temporally dispersed with a mean interval of 2.5 ms can still evoke supralinear spine calcium elevations. Although LTP induced by supralinear spine calcium elevation in response to coordinated synaptic inputs has not been experimentally demonstrated in the striatum, results from our model along with findings from other brain regions predict that striatal SPNs will exhibit cooperative LTP, and that strict spatial clustering and precise temporal synchrony of synaptic inputs may not be required.

Our finding that non-stimulated neighboring spines exhibit intermediate levels of calcium elevation suggests that these spines may undergo heterosynaptic plasticity (potentiation or depression) or metaplasticity. For instance, synaptic stimulation and induction of plasticity can induce metaplastic changes at nearby, non-stimulated spines in hippocampal pyramidal neurons (*Govindarajan et al., 2011*; *Harvey and Svoboda, 2007*). In contrast to metaplasticity, a recent hippocampal study found heterosynaptic depression of nearby non-stimulated spines when a cluster of synaptically stimulated spines underwent potentiation (*Oh et al., 2015*). Our results suggest that coordinated synaptic stimulation of clustered spines causes sufficient depolarization of neighboring non-stimulated spines for calcium elevation via VGCCs, which may underlie heterosynaptic changes (LTP or LTD) in nearby spines.

Calcium-dependent plasticity may also occur in the dendritic shaft. Dendritic spikes in pyramidal neurons can induce branch-specific potentiation of dendritic branch strength, via NMDAR-dependent regulation of dendritic potassium (Kv4.2) channels (*Losonczy et al., 2008*). Our results showing a moderate calcium elevation in the distal dendritic shaft suggest a mechanism linking NMDAR-dependent plateau potentials to dendritic branch strength plasticity. As the same (Kv4.2) potassium channels are expressed in SPN dendrites, we predict that branch-strength plasticity may also occur in SPNs.

Our most novel result is that inhibition enhances spatial specificity. Although inhibition also attenuates the supralinear response to coordinated synaptic inputs, the reduction in non-stimulated spines is more significant. Our finding that inhibitory synaptic input located distally on the dendritic branch attenuates the supralinear response to coordinated excitatory synaptic inputs is consistent with recent studies in the striatum (*Du et al., 2017*) and in cortical pyramidal neurons (*Doron et al., 2017*). However, we additionally found that inhibition enhances the spatial specificity of spine calcium elevation in response to coordinated excitatory inputs, a novel finding with implications for calcium-dependent signaling pathways. The role of distal (but not proximal) inhibition in regulating dendritic and spine calcium transients is consistent with a study showing that distal inhibition regulates bAP-induced calcium influx in cortical pyramidal neurons (*Marlin and Carter, 2014*), and a study showing that distal inhibitory inputs exhibit stronger inhibition of excitatory potentials than proximal inhibitory inputs (*Gidon and Segev, 2012*). Our findings also are consistent with a model study showing that inhibition can regulate the direction of homosynaptic (input-specific) plasticity for dendritic shaft synapses (*Bar-Ilan et al., 2012*), but we additionally show that inhibition can regulate both heterosynaptic and homosynaptic calcium signaling in dendritic spines. Our results suggest that the complex functional roles of inhibition may include regulating the balance of LTP and LTD within the dendritic branch by, for instance, changing heterosynaptic LTP to LTD; heterosynaptic LTD to no change in synaptic strength; or limiting the spatial extent in the dendrite of heterosynaptic plasticity. In agreement with *Du et al. (2017)*, inhibition located distally on the dendritic branch (near the site of excitatory inputs) within a limited time window relative to excitatory inputs exerted the strongest inhibitory effect. These results suggest that interneurons targeting distal dendrites, such as LTSIs or neuropeptide Y-expressing NGFs, as well as SPN collaterals, may regulate cooperative LTP to support cell assembly formation in the striatum (*Ibáñez-Sandoval et al., 2011*; *Ponzi and Wickens, 2013*; *Straub et al., 2016*; *Tepper et al., 2010*).

Importantly, our results are robust to parameter variations. The enhancement of spatial specificity by inhibition is most sensitive to NMDAR, CaR, and GABAR conductances, consistent with the NMDAR and CaR channels being the major sources of depolarization and calcium influx on synaptically stimulated and non-stimulated spines, respectively. The main finding that inhibition enhances spatial specificity was observed for all parameter variations (except for a 20% decrease in NMDAR conductance), although the amount that inhibition enhanced spatial specificity did depend on parameter values. Interestingly, both increases and decreases in NMDAR and CaR conductance reduced the ability for inhibition to increase spatial specificity. In the case of increased NMDAR or CaR conductance, greater depolarization would spread into non-stimulated spines, and the ability of the inhibitory current to counter the depolarization would be reduced. In the case of decreased NMDAR or CaR conductance, synapse-specificity is increased in the control (no inhibition) condition because weaker depolarization leads to significantly lower calcium influx to non-stimulated spines; thus, the ability of inhibition to further enhance spatial specificity is reduced. The sensitivity to NMDAR and CaR conductances is consistent with these channels being critical for plateau potential generation and duration in SPNs (*Plotkin et al., 2011*), and suggests that an optimal balance of inward currents may be necessary for spatial specificity.

The degree to which inhibition enhances spatial-specificity also is quite sensitive to GABAR conductance; however, the qualitative outcome is consistent for the entire range of GABAR conductances we evaluated. Therefore, the strength of inhibitory synapses may critically regulate spatial specificity and synaptic plasticity. We predict that calcium-dependent plasticity of inhibitory synapses, as has been observed in pyramidal neurons (*Chiu et al., 2018*), may regulate the degree of spatial specificity in a branch-specific manner.

The computational modeling methods we employed assume that calcium dynamics in dendritic spines can be modeled deterministically with one-dimensional axial diffusion of calcium and mobile buffers. These assumptions may limit our ability to fully capture calcium dynamics in dendritic spines. However, the scope of the model, which includes the entire dendritic morphology with thousands of dendritic spines, precludes simulating stochastic reaction kinetics and diffusion in three-dimensions. Crucially, we verified that our conclusions did not depend on limitations of our modeling methodology. Specifically, our method may underestimate the calcium diffusion rate from narrow spine necks to the dendritic shaft (*Stiles et al., 1996*). However, neither varying the calcium diffusion rate nor facilitating spine-neck diffusion affected our main result that spine calcium transients exhibit spatial specificity which is enhanced by inhibition. Our examination of mechanisms underlying spatial

specificity showed that calcium permeable ligand-gated and voltage-gated ion channels on synaptically stimulated and non-stimulated spines, respectively, were the main contributors to the amplitude of the calcium transients, and that diffusion only had a limited effect on the duration of calcium elevation in non-stimulated spines.

The calcium modeling method of the GENESIS simulator also does not account for the local radius of curvature of the cell membrane, which may affect fluxes and reactions (i.e. calcium pumps) at the membrane (*Bell et al., 2018*; *Rangamani et al., 2013*). However, our main result—that inhibition enhances spatial specificity—is robust to variations in parameters affecting calcium influx and efflux. Thus, although the exact magnitude of calcium concentration we report may be limited in accuracy because of methodological limitations, the key conclusions based on the ratios of calcium concentration in stimulated versus non-stimulated spines are qualitatively unchanged. Our conclusions are also supported by the model's ability to reproduce observations from calcium imaging experiments for stimulation of individual spines. Further, the quantity of spine calcium that is free, bound, pumped or diffuses out during synaptic stimulation in our model exhibits similar dynamics to a detailed three-dimensional model of spine calcium dynamics (*Bartol et al., 2015*). However, as our model does not include variations in spine geometry, it would be of interest for future work to consider the effects of realistic spine geometries (including curvatures) and three-dimensional reaction-diffusion modeling on spine calcium transients during dendritic plateau potentials.

Together, our results have implications for striatal function and plasticity in vivo. With inputs likely to be spatiotemporally distributed, our results suggest that fewer than 10 spines per branch may produce supralinear spine calcium when a few hundred total synapses on the dendritic tree are stimulated, similar to estimates of the number of synaptic inputs driving striatal upstates (*Blackwell et al., 2003*). Inhibitory inputs in vivo may control stimulus-specific and heterosynaptic plasticity, serving as a homeostatic mechanism to balance potentiation of excitatory inputs on a dendritic branch-specific level. Our recent work has shown that a calcium-based plasticity rule reproduces diverse in vitro plasticity protocols (*Jędrzejewska-Szmek et al., 2017*). Qualitatively extending this calcium-based plasticity rule to our results using clustered synaptic inputs (using the same threshold values reported) would predict that, in the absence of inhibition, supralinear spine calcium transients would produce LTP in synaptically stimulated spines, but also that non-stimulated neighboring spines would be above the LTP threshold and undergo heterosynaptic LTP. In this scenario, inhibition could reduce spine calcium elevation in non-stimulated spines to be above the LTD threshold but below the LTP threshold, switching the response from heterosynaptic LTP to heterosynaptic LTD for many (although not all) non-stimulated spines, whereas the responses in synaptically stimulated spines would remain above the LTP threshold. In the more in vivo-like conditions we simulated with spatiotemporally distributed inputs (*Figure 7C–D*), the spine calcium values we observed are more consistent with the plasticity rule thresholds, predicting LTP in synaptically stimulated spines and LTP, LTD, or no change in non-stimulated spines, with inhibition switching heterosynaptic LTP to LTD for non-stimulated spines. Extending this calcium-based plasticity rule to assess plasticity during repeated in vivo-like inputs may yield key predictions about how spatiotemporally distributed in vivo activity with trial-to-trial variability induces stable, synapse-specific plasticity.

## Materials and methods

We developed a biologically detailed multicompartment SPN model to investigate the effects of spatiotemporal patterns of synaptic input on calcium signaling. The model includes characterized morphology of SPN dendrites with explicitly modeled spines and ion channels that have been identified in SPNs. Uniquely, the model also includes sophisticated calcium dynamics consisting of calcium buffers, membrane pumps, and radial diffusion in dendrites and spines that enables us to predict how synaptic integration affects SPN calcium signaling.

### SPN model morphology and passive membrane properties

A biophysically detailed SPN model we previously published (*Jędrzejewska-Szmek et al., 2017*) was modified for this study (*Figure 1A*). The morphology consisted of a single cylindrical soma (11.3 μm length, 22.6 μm diameter) with four primary dendrites (12 μm length, 2.25 μm diameter), each branching twice into a total of eight secondary dendrites (14 μm length, 1.4 μm diameter) and 16 tertiary dendrites (198 μm length, tapered diameter from 0.89 μm proximally to 0.3 μm diameter

distally [*Wilson, 1992*]). Tertiary dendritic branches were subdivided into 3 µm long compartments to accurately model interactions among neighboring dendritic spines (*Gulledge et al., 2012*). Spines were explicitly modeled as a cylindrical head (0.5 µm diameter, 0.5 µm length) and neck (0.12 µm diameter, 0.5 µm length) and were distributed on secondary and tertiary dendritic branches with a density of 1 spine/µm, for a total of 3280 spines in the entire model.

Membrane resistivity and capacitity were set to 1.875 ohms-m$^2$ and 0.01 Farads/m$^2$, respectively. Axial resistance was set to 1.25 ohm-m for all compartments except for spine neck compartments, which were set to 11.3 ohm-m to achieve a neck resistance of 500 MΩ, as estimated from experimental data (*Harnett et al., 2012*). Passive parameters were determined by fitting the model to hyperpolarizing current injection (*Figure 1B*).

## Voltage-gated ionic channels

As described previously (*Jędrzejewska-Szmek et al., 2017*), the model includes the following voltage-gated sodium and potassium ion channels (*Table 1*): A fast sodium channel (NaF) (*Ogata and Tatebayashi, 1990*); fast (Kaf/Kv4.2) (*Tkatch et al., 2000*) and slow (Kas/Kv1.2) (*Shen et al., 2004*) A-type potassium channels; an inwardly rectifying potassium channel (Kir) (*Steephen and Manchanda, 2009*); and a resistant persistent potassium channel (Krp) (*Nisenbaum and Wilson, 1995*). Additionally, the model includes a big conductance voltage- and calcium-activated potassium channel (BK) (*Berkefeld et al., 2006*) and a small conductance calcium-activated potassium channel (SK) (*Maylie et al., 2004*). Six VGCCs are also included in the model (*Table 1*): CaR (*Brevi et al., 2001*; *Foehring et al., 2000*), CaN (Cav2.2) (*Bargas et al., 1994*; *Kasai and Neher, 1992*; *McNaughton and Randall, 1997*), CaL1.2 (Cav1.2) (*Bargas et al., 1994*; *Kasai and Neher, 1992*; *Tuckwell, 2012*), CaT3.2 (Cav3.2/ α1H) (*McRory et al., 2001*), CaT3.3 (Cav3.3/ α1I) (*McRory et al., 2001*), and CaL1.3 (Cav1.3) (*Tuckwell, 2012*). Channel kinetic equations and parameters are similar to our previously reported model (*Jędrzejewska-Szmek et al., 2017*), except we converted the previously nonspecific CaT channel to CaT3.3 and added a CaT3.2 channel with the following parameters: $m\ v_{half}$ = −43.15 mV; $m\ v_{slope}$ = −5.43 mV; $h\ v_{half}$ = −73.9 mV; $h\ v_{slope}$ = 2.76 mV; $m$ tau alpha rate = 160,000/V/s; $m$ tau alpha $v_{half}$ = 112; $m$ tau alpha $v_{slope}$ = 11; $m$ tau beta rate = 8500; $m$ tau beta $v_{slope}$ = 12.5; $m$ tau baseline offset = 0.0009 s; $h_{tau} = 22.25 + 0.0455e^{\frac{-Vm\ (mV)}{7.46}}$(ms). Channel conductance values were tuned to reproduce electrophysiology recordings (*Table 1*). The soma and

**Table 1.** Voltage-gated ion channel maximal conductances and permeabilities.

| Gbar (S/m$^2$) | Soma | Prox dend | Mid dend | Dist dend | Spine |
|---|---|---|---|---|---|
| NaF | 45,000 | 4420 | 4420 | 0 | 0 |
| Kir | 11.9 | 5.95 | 5.95 | 5.95 | 0 |
| KaF | 500 | 500 | 72 | 72 | 0 |
| KaS | 70 | 3 | 3 | 3 | 0 |
| Krp | 10 | 1 | 1 | 1 | 0 |
| SK | 3 | 0.5 | 0.5 | 0.5 | 0.5 |
| BK | 5 | 2 | 2 | 2 | 0 |
| Pbar (cm/s) | | | | | |
| CaL1.2 | 1.5e-7 | 1.5e-7 | 1.5e-7 | 1.5e-7 | 0.915e-7 |
| CaL1.3 | 0.5e-7 | 0.25e-7 | 0.25e-7 | 0.25e-7 | 0.1525e-7 |
| CaN | 15e-7 | 0 | 0 | 0 | 0 |
| CaR | 3e-7 | 30e-7 | 30e-7 | 30e-7 | 18.67e-8 |
| CaT (3.2) | 0 | 1.2e-7 | 2e-7 | 2e-7 | 1.22e-7 |
| CaT (3.3) | 0 | 0 | 5e-10 | 5e-10 | 3.42e-10 |

Gbar = maximal conductance (S/m2); Pbar = maximal calcium permeability. Prox dend = proximal dendrites (0 to 42 µm from soma); mid dend = middle dendrites (42–60 µm from soma); dist dend = distal dendrites (60–224 µm from soma).

DOI: https://doi.org/10.7554/eLife.38588.015

dendrites contain NaF, Kaf, Kas, Krp, and BK channels; SK channels are present in the soma and dendritic spines (*Higley and Sabatini, 2010*).

## Calcium dynamics

Calcium currents are modeled with the Goldman-Hodgkin-Katz (GHK) current equation to accurately account for the calcium driving potential. Calcium-dependent inactivation (CDI) was implemented for CaR, CaN, CaL1.2, and CaL1.3 channels (*Liang et al., 2003*). CaT channels were located in spines and distal dendrites (*Carter and Sabatini, 2004*; *McRory et al., 2001*; *Plotkin et al., 2011*), but not soma or proximal dendrites (*Bargas et al., 1994*). CaR, CaL1.2, and CaL1.3 channels were located in soma, dendrites, and spines (*Carter and Sabatini, 2004*; *Higley and Sabatini, 2010*). CaN channels were restricted to the soma (*Carter and Sabatini, 2004*). Calcium channel densities were tuned to experimentally reported calcium imaging for synaptic activation of a single spine (*Higley and Sabatini, 2010*; *Shindou et al., 2011*) and back-propagating AP-induced calcium influx into dendrites (*Carter and Sabatini, 2004*; *Day et al., 2008*; *Kerr and Plenz, 2002*; *Shindou et al., 2011*) and spines (*Carter and Sabatini, 2004*; *Shindou et al., 2011*) (*Figure 1B*). Contribution of specific channels to calcium influx was tuned to experiments blocking specific channel types (*Carter and Sabatini, 2004*; *Higley and Sabatini, 2010*).

Intracellular calcium concentration, diffusion, buffers, and pumps were modeled with the *difshell* object in GENESIS (*Bower and Beeman, 1998*). Calcium had a diffusion constant of 200 μm²/s (*Allbritton et al., 1992*). One-dimensional radial diffusion was implemented in the dendrites and soma by subdividing each cylindrical electrical compartment into a series of concentric shells; the submembrane shell had a diameter of 0.1 μm, and successive shells doubled in diameter (*Anwar et al., 2014*). One-dimensional axial diffusion was modeled in the spines and necks by subdividing the spine and neck electrical compartment into six cylindrical slabs, three for each compartment. Diffusion was also implemented between the spine neck and the submembrane shell of the dendrite. Calcium extrusion was implemented with Michaelis-Menten models of a plasma membrane calcium ATPase (PMCA) in the soma, dendrites, and spines, and a sodium-calcium exchanger (NCX) in the spines (*Table 2*). Calcium-permeable ion channels provide calcium influx to the spine head slabs and the dendritic/somatic submembrane shell. In the spine head, CaL1.3 (*Olson et al., 2005*) and calcium-permeable synaptic channels provide calcium influx to the outermost calcium slab (the postsynaptic density), while CaL1.2, CaR, and CaT provide calcium influx to the middle slab. Additionally, the SK channel in spines was dependent on calcium concentration of the middle slab.

Calcium buffers (*Table 2*) were modeled with the *difbuffer* object in GENESIS, which allows for buffering of calcium within *difshells* and diffusion of buffers (calcium-bound or free) between

**Table 2.** Calcium dynamics parameters

| Pumps | $K_m$ (mM) | $K_{cat}$ Soma (pmol/cm²/s) | $K_{cat}$ Dend (pmol/cm²/s) | $K_{cat}$ Spine (pmol/cm²/s) |
|---|---|---|---|---|
| PMCA | 0.3e-3 | 85 | 10 | 0.6 |
| NCX | 1e-3 | - | - | 10 |
| | | | | |
| Buffers | $K_d$ (μM) | $K_f$ (/s/μM) | Quantity (μM) | Diff (m²/s) |
| Calbindin | 0.7e-3 | 28 | 80 | 66e-12 |
| CaMN | 0.01 | 100 | 15 | 66e-12 |
| CaMC | 1.5e-3 | 6 | 15 | 66e-12 |
| Fixed | 100 | 400 | 2500 | 0 |
| Fluo-5F | 2.3 | 236 | 300 | 60e-12 |
| Fluo-4F | 9.7 | 80 | 200 | 60e-12 |
| Fura-2 | 0.185 | 1000 | 100 | 60e-12 |

PMCA = plasma membrane Ca²⁺ ATPase; NCX = sodium calcium exchanger; Fixed = endogenous immobilized buffer; CaMN = calmodulin N terminal binding site; CaMC = calmodulin C terminal binding site. Exogenous buffers were only present when tuning to calcium imaging experiments, in which case mobile endogenous buffers were removed.

DOI: https://doi.org/10.7554/eLife.38588.016

*difshells*. The model included the endogenous mobile buffers calbindin and calmodulin (N and C terminals), as well as an endogenous immobile buffer that was required to avoid unrealistic calcium elevations (*Matthews et al., 2013*; *Matthews and Dietrich, 2015*). The endogenous buffer quantities (*Table 2*) give a buffer capacity ratio close to 90, which is consistent with experimental estimates of buffer capacity in SPN spines and dendrites (*Carter and Sabatini, 2004*). Exogenous calcium buffers (calcium indicator dyes) were included in simulations when tuning to experimental calcium-imaging data.

### Synaptic channels

NMDAR and AMPAR synaptic channels were included on the spine heads and contributed calcium to the outermost spine head *difshell*. The fractional calcium currents were 5% of the total NMDAR current (implemented with the GHK current equation) and 0.1 % of the total AMPAR current. The AMPAR/NMDAR maximal conductance ratio was set to 1.0, and the conductances were set to achieve a unitary somatic PSP of ~2 mV, similar to the uncaging evoked EPSPs in *Plotkin et al., 2011*. Calcium-dependent inactivation of the NMDAR channel was implemented based on equations in a published model (*Farinella et al., 2014*).

In simulations that included GABA$_A$ stimulation, GABA$_A$ synaptic channels were included on the dendritic shaft with a maximal conductance of 1.2 nS. GABA$_A$ kinetics were either fast, consistent with synapses from fast spiking interneurons, low-threshold spiking interneurons, or SPN collaterals (*Straub et al., 2016*), or slow, consistent with NPY-neurogliaform synapses (*Ibáñez-Sandoval et al., 2011*).

### Simulation and analysis

Simulations were done with various spatiotemporal patterns of synaptic input as described in the results. In cases with asynchronous stimulation, the order of spine stimulation was randomly assigned. For simulations with random temporal dispersion, the ISI consisted of exponentially distributed intervals with an average ISI of 2.5, 5, or 10 ms per branch; for example the actual ISI for the 10 ms per branch case with two total branches stimulated was 5 ms, and with four total branches stimulated was 2.5 ms. This was done to make the total time of stimulation independent of number of branches stimulated, and to facilitate comparisons between simulations on a per-branches-stimulated basis (i.e. *Figure 4*). The minimum ISI values drawn from exponential distributions were unconstrained. The temporal order of asynchronously stimulated spines was randomly selected and followed no spatial pattern.

The model was simulated in GENESIS (*Bower and Beeman, 1998*) with a timestep of 0.01 ms. For simulations evaluating discretization, the timestep was increased to 0.001 ms, dendritic compartments were subdivided to 1 (rather than the standard 3) micron length compartments, the number of calcium shells in dendrites was increased using a constant shell depth equal to the submembrane shell depth, and calcium slabs in the spine head and neck were increased from three to six slabs. For simulations evaluating the effect of a coupling surface area from the spine head to neck or the spine neck to dendritic shaft, the GENESIS source code was changed to calculate the surface area for diffusion between two difshells (or difbuffers) using the geometric average of the surface areas of the two shells, rather than the GENESIS default of the minimum of the two shell surface areas.

Analysis was done in Python 2.7, using the Numpy, SciPy, Pandas, and Matplotlib python packages. Model simulation and analysis files are available on ModelDB. For statistical analysis of simulus specificity in *Figure 7C–D*, a cluster analysis was performed in SAS9.4, using duration and peak calcium as parameters to quantify the extent to which inhibition increased separation between stimulated and non-stimulated spines. The procedure FASTCLUS was used to create two clusters, and the output gave a measure of the distance between clusters of stimulated and non-stimulated spines. In addition, the procedure FREQ was applied to the output of the cluster analysis to generate the confusion matrices and identify the number of incorrectly labeled spines.

## Acknowledgements

This work was supported through the joint NIH-NSF CRCNS program through NIDA grant R01DA033390 and NIAAA grant R01AA16022

## Additional information

### Funding

| Funder | Grant reference number | Author |
|---|---|---|
| National Institute on Drug Abuse | R01DA033390 | Kim T Blackwell |
| National Institute on Alcohol Abuse and Alcoholism | R01AA16022 | Kim T Blackwell |

The funders had no role in study design, data collection and interpretation, or the decision to submit the work for publication.

### Author contributions

Daniel B Dorman, Conceptualization, Data curation, Software, Formal analysis, Investigation, Methodology, Writing—original draft, Writing—review and editing; Joanna Jędrzejewska-Szmek, Conceptualization, Software, Investigation, Methodology, Writing—review and editing; Kim T Blackwell, Conceptualization, Resources, Software, Supervision, Funding acquisition, Methodology, Project administration, Writing—review and editing

### Author ORCIDs

Daniel B Dorman (iD) http://orcid.org/0000-0001-7006-4593
Joanna Jędrzejewska-Szmek (iD) http://orcid.org/0000-0002-2336-0848
Kim T Blackwell (iD) https://orcid.org/0000-0003-4711-2344

### Decision letter and Author response

Decision letter https://doi.org/10.7554/eLife.38588.021
Author response https://doi.org/10.7554/eLife.38588.022

## Additional files

### Supplementary files

• Transparent reporting form
DOI: https://doi.org/10.7554/eLife.38588.017

### Data availability

All model simulation and analysis code are publicly and freely available on ModelDB (http://senselab.med.yale.edu/ModelDB/showModel.cshtml?model=245411).

The following dataset was generated:

| Author(s) | Year | Dataset title | Dataset URL | Database and Identifier |
|---|---|---|---|---|
| Daniel B Dorman, Joanna Jędrzejewska-Szmek, Kim T Blackwell | 2018 | Model simulation and analysis code | https://senselab.med.yale.edu/modeldb/enterCode.cshtml?model=245411 | ModelDB, 245411 |

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
