## [Decision Letter]

Thank you for submitting your article "Inhibition enhances spine-specific calcium encoding of synaptic input patterns in a biologically constrained model" for consideration by *eLife*. Your article has been reviewed by three peer reviewers, including Mary B Kennedy as the Reviewing Editor and Reviewer #1, and the evaluation has been overseen by Eve Marder as the Senior Editor. The following individuals involved in review of your submission have agreed to reveal their identity: Thomas M Bartol (Reviewer #2); Joshua Plotkin (Reviewer #3).

The reviewers have discussed the reviews with one another and the Reviewing Editor has drafted this decision to help you prepare a revised submission.

Summary:

The manuscript by Dorman et al. describes a biologically grounded compartmental model of a striatal spiny projection neuron developed with the program GENESIS. The authors use the model to examine dendritic synaptic integration and calcium dynamics. The model incorporates channels responsible for the experimentally observed electrical behavior of the neurons. Numbers and locations of membrane channels and pumps have been "tuned" to reproduce observed electrical behavior, as well as observed calcium transients in activated spines. The model includes both excitatory and inhibitory synaptic "inputs".

The authors verify that activation of a cluster of distal, but not proximal, dendritic spines produces a non-linear dendritic plateau potential, as observed experimentally. They then extend these findings to make testable predictions about synaptic cooperativity and heterosynaptic calcium dynamics in stimulated vs. unstimulated dendritic spines. They explore the effects of simultaneous vs. non-simultaneous coordinated stimulation of groups of distal spines on individual dendritic branches. They report that simultaneous coordinated synaptic stimulation evokes enhanced ("supralinear") elevation of calcium in stimulated spines and increases the specificity of the increase for stimulated vs. non-stimulated spines. This effect is specific to distal dendritic spines, suggesting that co-activation of spines on individual branches may lead to the most potent potentiation, and that distal branches may therefore act as relatively independent integrating units.

They also explore the effect of coordinated distal synaptic stimulation onto different, neighboring dendritic branches, finding that the effects of coordinate activation are less pronounced when the inputs are on different branches. The data offer several new insights into these processes, such as the effects of spatial and temporal dispersion of stimulation on calcium summation and the contribution of NMDA receptors vs. VGCCs in mediating spine-specific supralinear calcium responses.

The most novel finding, however, concerns the role of GABAergic inputs in shaping synaptic specificity during multi-synaptic stimulation. The authors show that single GABAergic synaptic inputs to distal dendrites can drastically increase the specificity of intra-spine calcium dynamics by increasing the difference between the sizes of calcium transients in stimulated vs. neighboring unstimulated spines. This has important implications for synaptic plasticity, and helps resolve a paradox of how synapse-specificity may be achieved under conditions where converging synaptic stimulation pushes an entire dendritic region into a depolarized plateau, a scenario that is now appreciated to represent a physiologically meaningful and important event.

Essential revisions:

The work represents the kind of modeling that the reviewers would like to see published in *eLife*. However, there are major concerns that must be addressed before the manuscript would be suitable for publication. The concerns fall into two categories; the first is the writing style which is, at present, inadequate for the relatively general readership of *eLife*. The second concern derives from limitations of the model structure that the authors need to address.

Writing:

1) The manuscript is poorly written and must be thoroughly re-written in a less terse and more didactic style to be suitable for the relatively broad audience of *eLife*. The writing becomes less and less clear as it progresses through the presentation of figures. Here are a few examples:

The title of the section “Cooperative spine calcium exhibits dendritic branch independence” is vague. What is cooperative about the spine calcium? And what is independent about the branches? This ambiguous use of language is present throughout the writing in the paragraphs that follow. A better title would be something like "The supralinear calcium transient in spines that are activated simultaneously is largest when the co-activated spines are on the same dendritic branch."

In the following paragraph, the authors begin using the term "inputs" to mean either synapses themselves or activated synapses. The authors should use the term activation whenever they refer to activated synapses. For example, "To investigate the effect of spatial dispersion of inputs, synaptic inputs were randomly distributed…". This sentence is ambiguous.

It is not clear what the meaning is of "average interstimulus interval of 10 ms per branch with random temporal ordering". The methods do not describe exactly how this stimulation was structured. This leads to several questions about the stimulus itself:

If the "temporal ordering" was random, were there still a certain percentage of spines that were activated simultaneously?

2) The timing of the simulated FSI spike train relative to glutamatergic synaptic stimulation (Figure 6C) should be clarified.

3) The peak spine calcium data shown in Figure 6C is taken only from stimulated spines, correct? What length of dendrite was stimulated (with glutamatergic synapses) for each branch condition tested (1 vs. 2 branches), and was there a distance dependence of the simulated FSI train on the synaptic calcium responses (i.e. location of glutamatergic synapse on the branch, with location of FSI inputs fixed)?

4) In the experiments where distributed non-simultaneous synaptic stimulation was used to induce plateau potentials (e.g. Figure 4), is there a direction dependence (towards vs. away from soma) to stimulation, as shown for other neuron types? If so, does direction affect membrane potential, calcium dynamics or both?

There are many instances of use of jargon and/or ambiguous wording, as well as sentences that are too terse to be clear. The manuscript needs a thorough editing to employ a better organized and more didactic style. One more example, "the duration of proximal spine calcium was prolonged when the proximal spine was stimulated prior to the distal cluster." The authors mean – "the calcium transients in the proximal spines were prolonged when proximal spines were stimulated prior to..." Note that calcium does not have a duration.

The section on the effects of GABA inputs was very difficult to read for all the above reasons. The authors seem to use GABA as a code for "activation of GABAergic inputs". This is an example of lab jargon. The sixth paragraph of the subsection “GABA attenuates stimulated spine calcium but enhances spatial specificity” needs to be re-organized and each individual issue should be discussed in a separate paragraph (for example fast vs. slow GABAergic receptor kinetics.) Briefly define the nature and meaning of a "confusion matrix" for *eLife*'s more general readership.

In the Introduction, and throughout the Results section, the work should be better set into the context of previous work. This will not detract from the significance of the present work, rather it will enhance it. The supralinear response of a calcium transient in spines resulting from activation of NMDA receptors has long been studied in cortical and hippocampal synapses. Earlier hippocampal/cortical papers should be cited; for example, Schiller, Schiller and Clapham, 1998. A good early review is Sjöström and Nelson, 2002. A stochastic computer model of a hippocampal neuron that includes differential contributions to spine calcium from various sources is Bartol et al., 2015.

The Introduction should contain a separate paragraph discussing the key differences between the anatomy and physiology of striatal spiny projection neurons and hippocampal and cortical projection neurons, including the observed synaptic plateau potentials, the up-state and down-state, and the observed differences between proximal and distal spines. In the Introduction and/or Discussion, a separate paragraph on the potential roles of the different sources of inhibitory inputs should be included. The present organization of the manuscript includes this background material interspersed throughout the Introduction, Results and Discussion, which becomes confusing.

Model Structure:

1) GENESIS (and also a recent module added to Neuron) treat the problem of reaction/diffusion between compartments connected by bottlenecks (such as the spine neck connected to the dendritic shaft) as a 1D problem (i.e. diffusion along the axial dimension of a cylinder). It has been shown previously that this treatment greatly underestimates the diffusion flux across the mouth of the bottleneck (see Stiles et al., 1996). The underestimate is due to the formation of a 3D concentration gradient in the neighborhood of the mouth as the diffusing particles enter the larger space. This 3D diffusion gradient greatly increases the apparent flux area of the mouth. Only a full 3D method that captures this fine structure provides an accurate quantitative estimate of diffusion as it occurs in the space and time scales of real dendrites and spines. Because major conclusions of this study concern fluxes and exchange of calcium among clusters of spines on the dendrite, getting the diffusion rate right may be of critical importance to the conclusions. The authors should discuss this aspect of the model explicitly and, at the very least, include information about the sensitivity of the conclusion of the model to the diffusion rate of calcium out of spines into the dendrite. For example, if the diffusion rate were faster, as described in the Stiles paper, would the tuning of channel numbers have to be adjusted to reproduce experimental data? And would the major conclusions of the study regarding size and specificity of calcium fluxes in spines still hold?

2) In computational science and numerical methods, the choice of temporal and spatial discretization of a PDE model is critical to achieve physically accurate results. For this reason, it is customary to validate a model by a test of convergence at ever finer space and time scales and through cross-validation with other simulation methods to demonstrate a robust solution. The authors should test the validity of their model using one or more of these techniques.

3) The boundary conditions in PDE models of reaction/diffusion are also a critical factor to get right. It is very important to account for the localradius of curvature at the boundary in the diffusion and flux terms at the boundary. It is not clear that the PDEs employed in the difshell module of GENESIS account for this curvature. See equation 3 of Rangamani et al., 2013. This could have a significant impact on the inward Ca flux and outward pump fluxes.

4) After addressing points 1-3 the calcium dynamics in the spines and dendrite may very well be different and will require the authors to re-tune the tunable parameters to compensate and restore a good fit to the experimental observations. It is unclear whether the authors' conclusions will hold up or be weakened.

---

## [Author Response]

Essential revisionsThe work represents the kind of modeling that the reviewers would like to see published in eLife. However, there are major concerns that must be addressed before the manuscript would be suitable for publication. The concerns fall into two categories; the first is the writing style which is, at present, inadequate for the relatively general readership of eLife. The second concern derives from limitations of the model structure that the authors need to address.Writing:1) The manuscript is poorly written and must be thoroughly re-written in a less terse and more didactic style to be suitable for the relatively broad audience of eLife. The writing becomes less and less clear as it progresses through the presentation of figures.

We have thoroughly revised the manuscript in a more didactic style. We addressed each of the specific concerns raised in the review detailed below, and we also revised the entire manuscript to make it clearer throughout to address the broader concerns about the writing style. We solicited feedback from new PhD students in the lab and lay people to ensure that the manuscript is accessible to a broad audience.

The title of the section “Cooperative spine calcium exhibits dendritic branch independence” is vague. What is cooperative about the spine calcium? And what is independent about the branches? This ambiguous use of language is present throughout the writing in the paragraphs that follow. A better title would be something like "The supralinear calcium transient in spines that are activated simultaneously is largest when the co-activated spines are on the same dendritic branch."

We have revised the section title similar to that recommended, i.e. “Synaptic cooperativity is strongest for synaptic inputs located on the same dendritic branch”. We eliminated the phrase “cooperative spine calcium” and other jargon from the entire manuscript.

In the following paragraph, the authors begin using the term "inputs" to mean either synapses themselves or activated synapses. The authors should use the term activation whenever they refer to activated synapses. For example, "To investigate the effect of spatial dispersion of inputs, synaptic inputs were randomly distributed…". This sentence is ambiguous.

We have revised our use of the term “inputs” to only mean stimulated synapses throughout the manuscript (e.g. “synaptic inputs”, “stimulated synapses”, or “stimulated spines” to refer to synaptically stimulated spines), and we use the term “synapses” without “inputs” to refer to any synapses themselves throughout the manuscript.

It is not clear what the meaning is of "average interstimulus interval of 10 ms per branch with random temporal ordering". The methods do not describe exactly how this stimulation was structured. This leads to several questions about the stimulus itself.

We have clarified the phrase in this paragraph, which now states “we used both simultaneous synaptic inputs and temporal dispersion, created by using random, exponentially distributed intervals. Simulations used average interstimulus intervals (ISIs) of 2.5, 5, or 10 ms, between synaptic inputs on each branch (e.g. with a 10 ms mean ISI per branch, when 2 branches are stimulated, the overall mean ISI is 5 ms for all inputs) (Figure 4). Specifying the ISI per branch has the advantage that the total duration of synaptic stimulation is independent of the number of branches.” We have also added a more detailed description of the temporal pattern in the Materials and methods (subsection “Simulation and Analysis”, first paragraph).

If the "temporal ordering" was random, were there still a certain percentage of spines that were activated simultaneously?

Interstimulus intervals were randomly selected from an exponential distribution with the specified mean value. The minimum value was unconstrained, so it was possible for some intervals to be near zero. We have clarified this in the Materials and methods (subsection “Simulation and Analysis”, first paragraph). Based on the exponential distribution, for an ISI of 5 ms, 5.6% of values are expected to be less than 0.3 ms, and for an ISI of 2.5 ms, 10.6% of values are expected to be less than 0.3 ms.

2) The timing of the simulated FSI spike train relative to glutamatergic synaptic stimulation (Figure 6C) should be clarified.

We have clarified the figure caption and the description in the Results section to specify that “The onset of the FSI input train and the glutamatergic stimulation was simultaneous.”

3) The peak spine calcium data shown in Figure 6C is taken only from stimulated spines, correct? What length of dendrite was stimulated (with glutamatergic synapses) for each branch condition tested (1 vs. 2 branches), and was there a distance dependence of the simulated FSI train on the synaptic calcium responses (i.e. location of glutamatergic synapse on the branch, with location of FSI inputs fixed)?

The referee is correct that peak spine calcium is shown only for stimulated spines in Figure 6C. Stimulated glutamatergic synapses were distributed over the entire dendritic branch (subsection “Inhibition attenuates calcium elevation in stimulated spines”, last paragraph). We have re-analyzed our results to determine whether the synapses that were facilitated are predominantly proximal or distal. We have added these results as Figure 6—figure supplement 1, and we added a description of these results in the Results section (see the aforementioned paragraph). This analysis revealed that the more distally located spines exhibited facilitated spine calcium responses (while proximal ones did not), indicating that the depolarizing caused by the FSI train (at the hyperpolarized SPN resting potential) propagates to the distal spines enough to facilitate calcium influx.

4) In the experiments where distributed non-simultaneous synaptic stimulation was used to induce plateau potentials (e.g. Figure 4), is there a direction dependence (towards vs. away from soma) to stimulation, as shown for other neuron types? If so, does direction affect membrane potential, calcium dynamics or both?

The temporal order for non-simultaneous synaptic stimulation was random and did not follow a spatial or distance-dependent rule, as we intended to simulate more in vivo like conditions for the simulations with spatiotemporally distributed inputs. Further, Plotkin et al. showed experimentally that there was no direction dependence of the somatic plateau potential when glutamatergic stimulation was applied in a distal-to-proximal or proximal-to-distal direction with 2 ms interstimulus intervals (Supplementary Figure 2 in Plotkin et al., 2011); thus, we did not investigate whether our model exhibited a direction dependence. We have revised the description of these simulations in the Results section to clarify that the spatiotemporally distributed inputs were applied without a spatial pattern (subsection “Synaptic cooperativity is strongest for synaptic inputs located on the same dendritic branch”, end of first paragraph), and also in the Materials and methods (subsection “Simulation and Analysis”, first paragraph).

There are many instances of use of jargon and/or ambiguous wording, as well as sentences that are too terse to be clear. The manuscript needs a thorough editing to employ a better organized and more didactic style. One more example, "the duration of proximal spine calcium was prolonged when the proximal spine was stimulated prior to the distal cluster." The authors mean – "the calcium transients in the proximal spines were prolonged when proximal spines were stimulated prior to…" Note that calcium does not have a duration.

We have thoroughly edited the manuscript throughout to consistently refer to “calcium elevation”, “calcium response”, or “calcium transients”, rather than the shorthand, ambiguously used “calcium”. We have specifically revised the sentence the reviewer gave as an example to state: “the duration of the calcium transient in the proximal spine was prolonged when the proximal spine was stimulated prior to the distal cluster”.

The section on the effects of GABA inputs was very difficult to read for all the above reasons. The authors seem to use GABA as a code for "activation of GABAergic inputs". This is an example of lab jargon. The sixth paragraph of the subsection “GABA attenuates stimulated spine calcium but enhances spatial specificity” needs to be re-organized and each individual issue should be discussed in a separate paragraph (for example fast vs. slow GABAergic receptor kinetics.)

We have revised our use of “GABA” to use “inhibition”, “GABAergic”, or “stimulation of GABAergic synapses” consistently throughout. We extensively revised the section on inhibition, including breaking it into 2 distinct sections: “Inhibition attenuates calcium elevation in stimulated spines”, and “Inhibition enhances stimulus-specificity of spine calcium transients”. However, we elected not to put fast vs. slow GABAergic receptor kinetics into separate paragraphs, as these results were qualitatively similar and only a single sentence is needed to state the results using fast GABAergic receptor kinetics once the results with slow GABAergic receptor kinetics are explained. Nonetheless, we revised our description of these results to make them clearer, and we maintained parallel sentence structure in descriptions of the fast and slow kinetics in each of the separate paragraphs on the timing or the location of GABAergic inputs (subsection “Inhibition attenuates calcium elevation in stimulated spines”, second and third paragraphs).

Briefly define the nature and meaning of a "confusion matrix" for eLife's more general readership.

We have added a definition and description of the confusion matrix in the Results (subsection “Inhibition enhances stimulus-specificity of spine calcium transients”, last paragraph).

In the Introduction, and throughout the Results section, the work should be better set into the context of previous work. This will not detract from the significance of the present work, rather it will enhance it. The supralinear response of a calcium transient in spines resulting from activation of NMDA receptors has long been studied in cortical and hippocampal synapses. Earlier hippocampal/cortical papers should be cited; for example, Schiller, Schiller and Clapham, 1998. A good early review is Sjöström and Nelson, 2002. A stochastic computer model of a hippocampal neuron that includes differential contributions to spine calcium from various sources is Bartol et al., 2015.

The Introduction, Results, and Discussion has been extensively revised to include greater context of previous work. We reordered the Introduction to provide better flow and to better connect our work with the literature. We have included the suggested references in our paragraphs in the Introduction on non-linear forms of synaptic integration (second paragraph)and on supralinear plateau potentials producing elevated dendritic calcium concentration(fourth paragraph).In addition,we further discuss Bartol et al., 2015 in the Results and Discussion, as we found this paper specifically insightful for its presentation of calcium fates in dendritic spines (i.e. Figure 9 in Bartol et al.). Thus, we performed an additional analysis and added a supplementary figure to our manuscript (Figure 1—figure supplement 2) with a similar presentation of calcium fates, for comparison to Bartol et al.

The Introduction should contain a separate paragraph discussing the key differences between the anatomy and physiology of striatal spiny projection neurons and hippocampal and cortical projection neurons, including the observed synaptic plateau potentials, the up-state and down-state, and the observed differences between proximal and distal spines. In the Introduction and/or Discussion, a separate paragraph on the potential roles of the different sources of inhibitory inputs should be included. The present organization of the manuscript includes this background material interspersed throughout the Introduction, Results and Discussion, which becomes confusing.

We have added a paragraph to the Introduction on key differences between striatal spiny projection neurons and pyramidal neurons (sixth paragraph), and a paragraph on the roles of inhibitory sources in the striatum (seventh paragraph). We have also extended comparisons to published work from pyramidal neurons in the Discussion (e.g. third paragraph).

Model Structure:1) […] The underestimate is due to the formation of a 3D concentration gradient in the neighborhood of the mouth as the diffusing particles enter the larger space. This 3D diffusion gradient greatly increases the apparent flux area of the mouth. Only a full 3D method that captures this fine structure provides an accurate quantitative estimate of diffusion as it occurs in the space and time scales of real dendrites and spines. Because major conclusions of this study concern fluxes and exchange of calcium among clusters of spines on the dendrite, getting the diffusion rate right may be of critical importance to the conclusions. The authors should discuss this aspect of the model explicitly and, at the very least, include information about the sensitivity of the conclusion of the model to the diffusion rate of calcium out of spines into the dendrite. For example, if the diffusion rate were faster, as described in the Stiles paper, would the tuning of channel numbers have to be adjusted to reproduce experimental data? And would the major conclusions of the study regarding size and specificity of calcium fluxes in spines still hold?

We are grateful for the reviewers’ comments addressing the model structure. We acknowledge that the one-dimensional approximation for diffusion and reaction is an approximation that should be more fully explored. The scope of our model, which includes the entire dendritic morphology and explicitly includes 3,280 dendritic spines, precludes a full 3D representation of reaction and diffusion. However, it is critical that our conclusions do not depend on the limitation of simulating calcium dynamics with a 1D approximation. To address the sensitivity of our conclusions that spine calcium transients exhibit stimulus-specificity and that inhibition enhances this specificity to the calcium diffusion parameters, we performed several simulations addressed here and in response to the subsequent specific items of concerns raised by the reviewers.

First, we conducted additional sensitivity analysis simulations while varying the calcium diffusion rate from -50% to +100% (Figure 7—figure supplement 2) and observed no qualitative change in the outcome of our main result. We have added a description in the Results explaining our analysis of model robustness and sensitivity to diffusion parameters (subsection “Inhibition enhances stimulus-specificity of spine calcium transients”, fourth paragraph) and added paragraphs in the Discussion (ninth and tenth paragraphs) on the limitations of modeling 1D, deterministic calcium diffusion and reactions. Similar to what we showed with our initial sensitivity analysis (Figure 7—figure supplement 1) and our analysis of mechanisms underlying synapse-specific calcium transients (Figure 3), these additional results confirm that the specificity of spine calcium responses most strongly depends on distinct sources of calcium influx, and that diffusion of calcium only has a small effect.

To further explore diffusion rate between the neck and dendritic shaft (or spine head), we modified the GENESIS source code to replace the default coupling coefficient for the shared surface area, which was the smaller of the two surface areas, with the geometric average of the two surface areas, which facilitates diffusion from the spine neck to the dendrite (Holmes, 2005) This result has been included in the additional sensitivity analysis figure (Figure 7—figure supplement 2, subsection “Inhibition enhances stimulus-specificity of spine calcium transients”, fourth paragraph), showing no qualitative effect on the main result.

2) In computational science and numerical methods, the choice of temporal and spatial discretization of a PDE model is critical to achieve physically accurate results. For this reason, it is customary to validate a model by a test of convergence at ever finer space and time scales and through cross-validation with other simulation methods to demonstrate a robust solution. The authors should test the validity of their model using one or more of these techniques.

We have addressed the concern about temporal and spatial discretization by simulating the model with finer temporal and spatial scales. We decreased the simulation time step from 10 to 1 microsecond; we decreased the voltage compartment length in tertiary dendrites from 3 microns to 1 micron; and we increased the number of diffusion shells in the dendritic shafts, spine necks, and spine heads. We verified that our discretization of diffusion met stability requirements given by the equation D∆t(∆x)2≤12.

This finer discretization had a trivial effect on the outcome of membrane potential in the soma or a dendritic spine and spine calcium transients in stimulated or non-stimulated spines. Finer discretization did not qualitatively change the main result that inhibition enhances spatial specificity of spine calcium elevation (see Figure 7—figure supplement 2, subsection “Inhibition enhances stimulus-specificity of spine calcium transients”, fourth paragraph).

3) The boundary conditions in PDE models of reaction/diffusion are also a critical factor to get right. It is very important to account for the localradius of curvature at the boundary in the diffusion and flux terms at the boundary. It is not clear that the PDEs employed in the difshell module of GENESIS account for this curvature. See equation 3 of Rangamani et al., 2013. This could have a significant impact on the inward Ca flux and outward pump fluxes.

It is true that the difshell object in GENESIS does not account for local radius of curvature, but rather approximates each discretized calcium shell as a well-mixed volume, and the fluxes at the membrane are approximated using the total difshell membrane surface area and calcium concentration in the submembrane volume. While the scope of our model precludes simulating calcium dynamics at the same scale as Rangamani et al., 2013, we have assessed the sensitivity of our results to inward calcium flux and outward pump flux, shown by Rangamani et al. to be influenced by local curvature. We performed simulations that varied inward calcium flux through calcium permeable channels (while not varying the contribution of these channels to membrane depolarization) and calcium pump density from -50% to +100%. These results, which have been added to the new sensitivity analysis figure (Figure 7—figure supplement 2, subsection “Inhibition enhances stimulus-specificity of spine calcium transients”, fourth paragraph) indicate that our main results are not qualitatively affected by the specific membrane calcium fluxes. Indeed, although the magnitudes of calcium transients depend strongly on these parameters, the role of inhibition in enhancing spine-specificity does not depend strongly on the absolute magnitudes of calcium fluxes. With as much as a 50% reduction in calcium influx, inhibition still enhanced spatial specificity, though the degree to which inhibition did so was reduced.

Additionally, to further address the limitations of the 1D approximation of calcium dynamics in our model, we have compared the fate of calcium influx into a spine in our model during a single synaptic stimulation (i.e. the amount of free, bound, pumped, or diffused calcium) with a published computational model incorporating 3D, stochastic calcium reaction-diffusion in reconstructed dendritic spines (Bartol et al., 2015) (subsection “Multiscale model reproduces electrophysiology and calcium-imaging experiments”, last paragraph). We show the fraction of buffered, free, diffused, and pumped calcium in response to a single synaptic stimulation (Figure 1—figure supplement 2), which is similar to Figure 9 in Bartol et al., 2015. We have addressed the limitations regarding calcium influx and efflux (and their dependence on membrane curvature) in the Discussion (tenth paragraph).

4) After addressing points 1-3 the calcium dynamics in the spines and dendrite may very well be different and will require the authors to re-tune the tunable parameters to compensate and restore a good fit to the experimental observations. It is unclear whether the authors' conclusions will hold up or be weakened.

As our main results were robust to the additional simulations we performed (presented in Figure 1—figure supplement 1, Figure 1—figure supplement 2, and Figure 7—figure supplement 2), we concluded that retuning the model was not necessary.